# BIDIRECTIONAL CONSISTENCY MODELS

## ABSTRACT

Diffusion models (DMs) are capable of generating remarkably high-quality samples by iteratively denoising a random vector, a process that corresponds to moving along the probability flow ordinary differential equation (PF ODE). Interestingly, DMs can also invert an input image to noise by moving backward along the PF ODE, a key operation for downstream tasks such as interpolation and image editing. However, the iterative nature of this process restricts its speed, hindering its broader application. Recently, Consistency Models (CMs) have emerged to address this challenge by approximating the integral of the PF ODE, largely reducing the number of iterations. Yet, the absence of an explicit ODE solver complicates the inversion process. To resolve this, we introduce Bidirectional Consistency Model (BCM), which learns a *single* neural network that enables both *forward and backward* traversal along the PF ODE, efficiently unifying generation and inversion tasks within one framework. We can train BCM from scratch or tune it using a pretrained consistency model, wh ich reduces the training cost and increases scalability. We demonstrate that BCM enables one-step generation and inversion while also allowing the use of additional steps to enhance generation quality or reduce reconstruction error. We further showcase BCM's capability in downstream tasks, such as interpolation, inpainting, and blind restoration of compressed images. Notably, when the number of function evaluations (NFE) is constrained, BCM surpasses domain-specific restoration methods, such as I$^2$SB and Palette, in a fully zero-shot manner, offering an efficient alternative for inversion problems.

## 1 INTRODUCTION

Two key components in image generation and manipulation are *generation* and its *inversion*. Specifically, *generation* aims to learn a mapping from simple noise distributions, such as Gaussian, to complex ones, like the distribution encompassing all real-world images. In contrast, *inversion* seeks to find the reverse mapping, transforming real data back into the corresponding noise [1]. Recent breakthroughs in deep generative models (Goodfellow et al., 2014; Kingma & Welling, 2013; Dinh et al., 2014; Song & Ermon, 2019; Ho et al., 2020; Song et al., 2021a;b) have revolutionized this field. These models have not only achieved remarkable success in synthesizing high-fidelity samples across various modalities (Karras et al., 2021; Rombach et al., 2022; Kong et al., 2021; OpenAI, 2024), but have proven effective in downstream applications, such as image editing, by leveraging the inversion process (Mokady et al., 2023; Huberman-Spiegelglas et al., 2023; Hertz et al., 2023).

Particularly, score-based diffusion models (DMs) (Song & Ermon, 2019; Ho et al., 2020; Song et al., 2021a;b; Karras et al., 2022) have stood out for generation (Dhariwal & Nichol, 2021). Starting from random initial noise, DMs progressively remove noise through a process akin to a numerical ODE solver operating over the probability flow ordinary differential equation (PF ODE), as outlined by Song et al. (2021b). However, it typically takes hundreds of iterations to produce high-quality generation results. This slow generation process limits broader applications.

This issue has been recently addressed by consistency models (CMs) (Song et al., 2023; Song & Dhariwal, 2024), which directly compute the integral of PF ODE trajectory from any time step to zero. Similar to CMs, Kim et al. (2024) introduced the Consistency Trajectory Model (CTM), which estimates the integral between any two time steps along the trajectory towards the denoising

---

[1]The inversion problem also refers to restoring a high-quality image from its degraded version. However, in this paper, we define it more narrowly as the task of finding the corresponding noise for an input image.

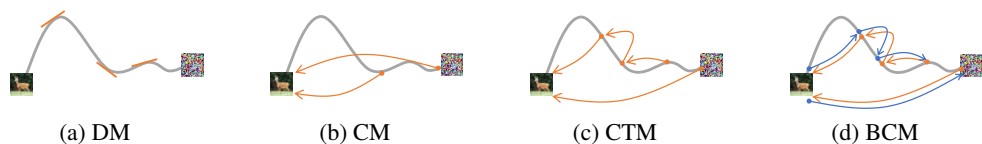

(a) DM        (b) CM        (c) CTM        (d) BCM

Figure 1: An illustrative comparison of score-based diffusion models, consistency models, consistency trajectory models, and our proposed bidirectional consistency models. (a) DM estimates the score function at a given time step; (b) CM enforces self-consistency that different points on the same trajectory map to the same initial points; (c) CTM strengthens this principle of consistency, which maps a point at time $t$ back to another point at time $u \leqslant t$ along the same trajectory. (d) BCM is designed to map any two points on the same trajectory to each other, removing any restrictions on the mapping direction. When the mapping direction aligns with the diffusion direction, the model adds noise to an input image. Conversely, if the mapping direction is opposite, the model performs denoising. This approach unifies generation and inversion tasks into a single, cohesive framework.

direction. Through these approaches, the consistency model family enables image generation with only a single Number of Function Evaluation (NFE, i.e., number of network forward passes) while offering a trade-off between speed and quality.

On the other hand, unfortunately, the inversion tasks remain challenging for DMs. First, the generation process in many DMs (Ho et al., 2020; Karras et al., 2022) is stochastic and hence is non-invertible; second, even for methods that employ a deterministic sampling process (Song et al., 2021a), inverting the ODE solver necessitates at least hundreds of iterations for a small reconstruction error; third, although CMs and CTM accelerate generation by learning the integral directly, this integration is strongly non-linear, making the inversion process even harder.

Therefore, in this work, we aim to bridge this gap through natural yet non-trivial extensions to CMs and CTM. Specifically, they possess a key feature of self-consistency: points along the same trajectory map back to the same initial point. Inspired by this property, we pose the questions:

*Is there a form of stronger consistency where points on the same trajectory can map to each other, regardless of their time steps' order? Can we train a network to learn this mapping?*

In the following sections of this paper, we affirmatively answer these questions with our proposed *Bidirectional Consistency Model* (BCM). Specifically:

1. We train a single neural network that enables both forward and backward traversal (i.e., integration) along the PF ODE, unifying the generation and inversion tasks into one framework. We can either train the model from scratch or fine-tune it from a pretrained consistency model to reduce training cost and increase scalability.

2. We demonstrate BCM can generate images or invert a given image with a single NFE, and can achieve improved sample quality or lower reconstruction error by chaining multiple time steps.

3. Leveraging BCM's capability to navigate both forward and backward along the PF ODE trajectory, we introduce two sampling schemes and a combined approach that has empirically demonstrated superior performance.

4. We apply BCM for image interpolation, inpainting, and blind restoration of compressed images, showcasing its potential versatile downstream tasks.

## 2 BACKGROUND AND PRELIMINARY

Before launching into the details of the Bidirectional Consistency Model (BCM), we first describe some preliminaries, including a brief introduction to Score-based Diffusion Models (DMs), Consistency Models (CMs), and Consistency Trajectory Model (CTM) in the following. We also illustrate these models, along with our proposed method, in Figure 1.

**Score-based Diffusion Models.** Score-based Diffusion Models (DMs, Song et al., 2021b) sample from the target data distribution by progressively removing noise from a random $\mathbf{x}_T \sim N(0, T^2 \boldsymbol{I})$.

To achieve this, DMs first diffuse the data distribution $p_{\text{data}}(\mathbf{x})$ through a stochastic differential equation (SDE): $d\mathbf{x}_t = \boldsymbol{\mu}(\mathbf{x}_t, t)dt + \sigma(t)d\mathbf{w}_t$, which has a reverse-time SDE, as described by Anderson (1982): $d\mathbf{x}_t = \left[ \boldsymbol{\mu}(\mathbf{x}_t, t) - \frac{1}{2}\sigma^2(t)\nabla \log p_t(\mathbf{x}_t) \right] dt + \sigma(t)d\bar{\mathbf{w}}_t$, where $t \in [0, T]^2$, $p_t$ is the marginal density of $\mathbf{x}_t$, and $\mathbf{w}_t$ and $\bar{\mathbf{w}}_t$ represents the standard Wiener process in forward and reverse time respectively. Note, that $p_0$ is our desired data distribution $p_{\text{data}}$. Remarkably, Song et al. (2021b) showed that there exists an ordinary differential equation, dubbed the Probability Flow (PF) ODE, whose solution trajectories have the same marginal density $p_t$ at time $t$:

$$d\mathbf{x}_t/dt = \boldsymbol{\mu}(\mathbf{x}_t, t) - 1/2\sigma^2(t)\nabla \log p_t(\mathbf{x}_t). \tag{1}$$

During training, DMs learn to estimate $\nabla \log p_t(\mathbf{x}_t)$ with a score model $\boldsymbol{s}(\mathbf{x}_t, t)$ with score matching (Hyvärinen & Dayan, 2005; Song et al., 2021b; Karras et al., 2022). And during sampling, DMs solve the empirical PF ODE from time $T$ to 0 with a numerical ODE solver. Following Karras et al. (2022), we set $\mu = 0, \sigma = \sqrt{2t}, T = 80$.

**Consistency Models, Consistency Training, and improved Consistency Training.** However, while the above procedure has proven effective in generating high-quality samples, the numerical ODE solver typically requires iterative evaluations of the network $\boldsymbol{s}(\mathbf{x}_t, t)$ and hence bottlenecks the generation speed. To this end, Song et al. (2023) proposed Consistency Models (CMs) that train a network to estimate the solution to PF ODE directly, i.e.,

$$\boldsymbol{f_\theta}(\mathbf{x}_t, t) \approx \mathbf{x}_0 = \mathbf{x}_t + \int_t^0 \left( d\mathbf{x}_s/ds \right) ds \tag{2}$$

The network $\boldsymbol{f_\theta}(\cdot, \cdot)$ can be either trained by distillation or from scratch with Consistency Training (CT). Here, we describe the consistency training in more detail since it lays the foundation of our proposed method: to begin, consistency training first discretizes the time horizon $[0, T]$ into $N - 1$ sub-intervals, with boundaries $0 = t_1 < t_2 < \cdots < t_N = T$. The training objective is defined as

$$\mathcal{L}_{CT}^N(\boldsymbol{\theta}, \boldsymbol{\theta}^-) = \mathbb{E}_{\mathbf{z},\mathbf{x},n}[\lambda(t_n)d(\boldsymbol{f_\theta}(\mathbf{x} + t_{n+1}\mathbf{z}, t_{n+1}), \boldsymbol{f_{\theta^-}}(\mathbf{x} + t_n\mathbf{z}, t_n))]. \tag{3}$$

$\boldsymbol{\theta}$ and $\boldsymbol{\theta}^-$ represents the parameters of the online network and a target network, respectively. The target network is obtained by $\boldsymbol{\theta}^- \leftarrow \texttt{stopgrad}(\mu\boldsymbol{\theta}^- + (1 - \mu)\boldsymbol{\theta})$ at each iteration. $\lambda(\cdot)$ is a reweighting function, $\mathbf{x}$ represents the training data sample, and $\mathbf{z} \sim \mathcal{N}(0, \boldsymbol{I})$ is a random Gaussian noise. During training, $N$ is gradually increased, allowing the model to learn self-consistency in an incremental manner. Additionally, Song et al. (2023) proposed to keep track of an exponential moving average (EMA) of the online parameter $\boldsymbol{\theta}$. Specifically, after optimizing $\boldsymbol{\theta}$ in each iteration, CMs update the EMA by $\boldsymbol{\theta}_{\text{EMA}} \leftarrow \mu_{\text{EMA}}\boldsymbol{\theta}_{\text{EMA}} + (1 - \mu_{\text{EMA}})\boldsymbol{\theta}$. After training, CMs discard the online parameters and generate samples with $\boldsymbol{\theta}_{\text{EMA}}$.

Besides, in a follow-up work, Song & Dhariwal (2024) suggested to use the Pseudo-Huber loss for $d$, along with other techniques that include setting $\mu = 0$ (i.e., $\boldsymbol{\theta}^- \leftarrow \texttt{stopgrad}(\boldsymbol{\theta})$), proposing a better scheduler function for $N$, adapting a better reweighting function $\lambda(t_n) = 1/|t_n - t_{n+1}|$. Dubbed improved Consistency Training (iCT), these modifications significantly improve the performance. Therefore, unless otherwise stated, we inherit these improving techniques in our work.

**Consistency Trajectory Model.** While CMs learn the integral from an arbitrary starting time to 0, Consistency Trajectory Model (CTM) (Kim et al., 2024) learns the integral between any two time steps along the PF ODE trajectory towards the denoising direction. More specifically, CTM learns

$$\boldsymbol{f_\theta}(\mathbf{x}_t, t, u) \approx \mathbf{x}_u = \mathbf{x}_t + \int_t^u \left( d\mathbf{x}_s/ds \right) ds, \quad \text{where } u \leqslant t. \tag{4}$$

CTM demonstrates that it is possible to learn a stronger consistency: two points $\mathbf{x}_t$ and $\mathbf{x}_u$ along the same trajectory not only can map back to the same initial point $\mathbf{x}_0$, but also can map from $\mathbf{x}_t$ to $\mathbf{x}_u$, provided $u \leqslant t$. This inspires us for a stronger consistency with a bijection between $\mathbf{x}_t$ and $\mathbf{x}_u$.

However, it is worth noting that, while sharing some motivation, our methodology is foundationally different from that in CTM. Our network adopts a different parameterization, which is a natural extension of that by EDM (Karras et al., 2022). Moreover, CTM largely relies on adversarial loss (Goodfellow et al., 2014), while our proposed method, following Song & Dhariwal (2024), is free from LPIPS (Zhang et al., 2018) and adversarial training.

---

[2]To avoid numerical issues, we always set $t$ in $[0.002, T]$ in practice. However, to keep the notation simple, we will ignore this small initial value 0.002 when describing our methods in this paper.

---

**Algorithm 1** Bidirectional Consistency Training (Red indicates differences from CT/iCT (Song et al., 2023; Song & Dhariwal, 2024)

---

**Input:** Training set $\mathcal{D}$, initial model parameter $\boldsymbol{\theta}$, learning rate $\eta$, step schedule $N(\cdot)$, noise schedule $p(\cdot)$, EMA rate $\mu_{\text{EMA}}$, distance metric $d(\cdot, \cdot)$, reweighting function $\lambda(\cdot)$ and $\lambda'(\cdot, \cdot)$.

**Output:** Model parameter $\boldsymbol{\theta}_{\text{EMA}}$.

    **Initialize:** $\boldsymbol{\theta}_{\text{EMA}} \leftarrow \boldsymbol{\theta}, k \leftarrow 0$.

    **repeat until convergence**

        # Sample training example, time steps, and random noise:

        Sample $\mathbf{x} \in \mathcal{D}$, $n \sim p(n|N(k))$.

        Sample $n' \sim \tilde{p}(n'|N(k))$, where $\tilde{p}(n'|N(k)) \propto \begin{cases} 0, & \text{if } n' = n, \\ p(n'|N(k)), & \text{otherwise.} \end{cases}$

        Sample $\mathbf{z} \sim \mathcal{N}(0, \boldsymbol{I})$.

        # Calculate and optimize BCT loss:

        $\mathcal{L}_{\text{CT}}(\boldsymbol{\theta}) \leftarrow \lambda(t_n) d(\boldsymbol{f}_{\boldsymbol{\theta}}(\mathbf{x} + t_{n+1}\mathbf{z}, t_{n+1}, 0), \boldsymbol{f}_{\bar{\boldsymbol{\theta}}}(\mathbf{x} + t_n\mathbf{z}, t_n, 0))$;

        $\mathcal{L}_{\text{ST}}(\boldsymbol{\theta}) \leftarrow \lambda'(t_n, t_{n'}) d(\boldsymbol{f}_{\bar{\boldsymbol{\theta}}}(\boldsymbol{f}_{\boldsymbol{\theta}}(\mathbf{x} + t_n\mathbf{z}, t_n, t_{n'}), t_{n'}, 0), \boldsymbol{f}_{\bar{\boldsymbol{\theta}}}(\mathbf{x} + t_n\mathbf{z}, t_n, 0))$;

        $\mathcal{L}_{\text{BCT}}(\boldsymbol{\theta}) \leftarrow \mathcal{L}_{\text{CT}}(\boldsymbol{\theta}) + \mathcal{L}_{\text{ST}}(\boldsymbol{\theta})$;

        $\boldsymbol{\theta} \leftarrow \boldsymbol{\theta} - \eta \nabla_{\boldsymbol{\theta}} \mathcal{L}_{\text{BCT}}(\boldsymbol{\theta})$;

        # Update the EMA parameter and the iteration number:

        $\boldsymbol{\theta}_{\text{EMA}} \leftarrow \mu_{\text{EMA}} \boldsymbol{\theta}_{\text{EMA}} + (1 - \mu_{\text{EMA}}) \boldsymbol{\theta}, k \leftarrow k + 1$;

    **end repeat**

---

## 3 METHODS

In this section, we describe details of Bidirectional Consistency Model (BCM). From a high-level perspective, we train a network $\boldsymbol{f}_{\boldsymbol{\theta}}(\mathbf{x}_t, t, u)$ that traverses along the probability flow (PF) ODE from time $t$ to time $u$, i.e., $\boldsymbol{f}_{\boldsymbol{\theta}}(\mathbf{x}_t, t, u) \approx \mathbf{x}_u = \mathbf{x}_t + \int_t^u \frac{d\mathbf{x}_s}{ds} ds$. This is similar to Equation (4), but since we aim to learn both generation and inversion, we do not set constraints on $t$ and $u$, except for $t \neq u$. To this end, we adjust the network parameterization and the training objective of consistency training (Song et al., 2023; Song & Dhariwal, 2024). These modifications are detailed in Sections 3.1 and 3.2. Besides, we introduce new sampling schemes leveraging our model's invertibility, which are described in Section 3.3. Finally, we discuss BCM's inversion in Section 3.4.

### 3.1 NETWORK PARAMETERIZATION

We first describe the network parameterization. Our network takes in three arguments: 1) the sample $\mathbf{x}_t$ at time $t$, 2) the current time step $t$, and 3) the target time step $u$, and outputs the sample at time $u$, i.e., $\mathbf{x}_u$. To achieve this, we directly expand the models used in Consistency Models (CMs) (Song et al., 2023; Song & Dhariwal, 2024) with an extra argument $u$. In CMs, the networks first calculate Fourier embeddings (Tancik et al., 2020) or positional embeddings (Vaswani et al., 2017) for the time step $t$, followed by two dense layers. Here, we simply concatenate the embeddings of $t$ and $u$, and double the dimensionality of the dense layers correspondingly.

Similar to CMs (Song et al., 2023) and EDM (Karras et al., 2022), instead of directly learning $\boldsymbol{f}_{\boldsymbol{\theta}}$, we train $\boldsymbol{F}_{\boldsymbol{\theta}}$ and let $\boldsymbol{f}_{\boldsymbol{\theta}}(\mathbf{x}_t, t, u) = c_{\text{skip}}(t, u)\mathbf{x}_t + c_{\text{out}}(t, u)F_{\boldsymbol{\theta}}(c_{\text{in}}(t, u)\mathbf{x}_t, t, u)$, where

$$c_{\text{in}}(t, u) = \frac{1}{\sqrt{\sigma_{\text{data}}^2 + t^2}}, \quad c_{\text{out}}(t, u) = \frac{\sigma_{\text{data}}(t - u)}{\sqrt{\sigma_{\text{data}}^2 + t^2}}, \quad c_{\text{skip}}(t, u) = \frac{\sigma_{\text{data}}^2 + tu}{\sigma_{\text{data}}^2 + t^2}. \quad (5)$$

Note that $c_{\text{skip}}(t, t) = 1$ and $c_{\text{out}}(t, t) = 0$, which explicitly enforce the boundary condition $\boldsymbol{f}_{\boldsymbol{\theta}}(\mathbf{x}_t, t, t) = \mathbf{x}_t$. We detail our motivation and derivations in Appendix D.

### 3.2 BIDIRECTIONAL CONSISTENCY TRAINING

We now discuss the training of BCM, which we dub as Bidirectional Consistency Training (BCT). Following Song et al. (2023); Song & Dhariwal (2024), we discretize the time horizon $[0, T]$ into $N - 1$ intervals with boundaries $0 = t_1 < t_2 < \cdots < t_N = T$.

Our training objective has two terms. The first term takes the same form as Equation (3), enforcing the consistency between any points on the trajectory and the starting point. We restate it with our

new parameterization for easier reference:

$$\mathcal{L}_{CT}^N(\boldsymbol{\theta}) = \mathbb{E}_{\mathbf{z},\mathbf{x},t_n}[\lambda(t_n)d(\boldsymbol{f_\theta}(\mathbf{x}+t_{n+1}\mathbf{z},t_{n+1},0),\boldsymbol{f_{\bar\theta}}(\mathbf{x}+t_n\mathbf{z},t_n,0))], \tag{6}$$

where $\mathbf{x}$ is one training sample, $\mathbf{z} \sim \mathcal{N}(0,\boldsymbol{I})$, and $\bar{\boldsymbol{\theta}}$ is the stop gradient operation on $\boldsymbol{\theta}$, and $\lambda(t_n) = {}^1/|t_n - t_{n+1}|$. We replace $\boldsymbol{\theta}^-$ in Equation (3) with $\bar{\boldsymbol{\theta}}$ according to Song & Dhariwal (2024).

The second term explicitly sets constraints between any two points on the trajectory. Specifically, given a training example $\mathbf{x}$, we randomly sample two time steps $t$ and $u$, and want to construct a mapping from $\mathbf{x}_t$ to $\mathbf{x}_u$, where $\mathbf{x}_t$ and $\mathbf{x}_u$ represent the results at time $t$ and $u$ along the Probability Flow (PF) ODE trajectory, respectively. Note, that the model learns to denoise (i.e., generate) when $u < t$, and to add noise (i.e., inverse) when $u > t$. Therefore, this single term unifies the generative and inverse tasks within one framework, and with more $t$ and $u$ sampled during training, we achieve consistency over the entire trajectory. To construct such a mapping, we minimize the distance

$$d\left(\boldsymbol{f_\theta}(\mathbf{x}_t,t,u),\mathbf{x}_u\right). \tag{7}$$

However, Equation (7) will have different scales for different $u$ values, leading to a high variance during training. Therefore, inspired by Kim et al. (2024), we map both $\boldsymbol{f_\theta}(\mathbf{x}_t,t,u)$ and $\mathbf{x}_u$ to time $0$, and minimize the distances between these two back-mapped images, i.e.,

$$d\left(\boldsymbol{f_{\bar\theta}}(\boldsymbol{f_\theta}(\mathbf{x}_t,t,u),u,0),\boldsymbol{f_{\bar\theta}}(\mathbf{x}_u,u,0)\right), \tag{8}$$

where $\bar{\boldsymbol{\theta}}$ is the same $\boldsymbol{\theta}$ with stop gradient operation. We denote this as a "soft" trajectory constraint. Unfortunately, directly minimizing Equation (8) without a pretrained DM is still problematic. This is because, without a pretrained DM, we can only generate $\mathbf{x}_t$ and $\mathbf{x}_u$ from the diffusion SDE, i.e., by adding Gaussian noise to $\mathbf{x}$. However, $\mathbf{x}_t$ and $\mathbf{x}_u$ generated by the diffusion SDE do not necessarily lie on the same PF ODE trajectory, and hence Equation (8) still fails to build the desired bidirectional consistency. Instead, we notice that when the CT loss defined in Equation (6) converges, we have $\boldsymbol{f_{\bar\theta}}(\mathbf{x}_u,u,0) \approx \boldsymbol{f_{\bar\theta}}(\mathbf{x}_t,t,0) \approx \mathbf{x}$. We therefore optimize:

$$d\left(\boldsymbol{f_{\bar\theta}}(\boldsymbol{f_\theta}(\mathbf{x}_t,t,u),u,0),\boldsymbol{f_{\bar\theta}}(\mathbf{x}_t,t,0)\right). \tag{9}$$

Empirically, we found Equation (9) plays a crucial role in ensuring accurate inversion performance. We provide experimental evidence for this loss choice in Appendix B.2.1.

Finally, we recognize that the term $\boldsymbol{f_{\bar\theta}}(\mathbf{x}+t_n\mathbf{z},t_n,0)$ in Equation (6) and the term $\boldsymbol{f_{\bar\theta}}(\mathbf{x}_t,t,0)$ in Equation (9) have exactly the same form. Therefore, we set $t = t_n$ to reduce one forward pass. Putting together, we define our objective as

$$\begin{aligned}\mathcal{L}_{BCT}^N(\boldsymbol{\theta}) = \mathbb{E}_{\mathbf{z},\mathbf{x},t_n,t_{n'}}\Big[&\lambda(t_n)d(\boldsymbol{f_\theta}(\mathbf{x}+t_{n+1}\mathbf{z},t_{n+1},0),\boldsymbol{f_{\bar\theta}}(\mathbf{x}+t_n\mathbf{z},t_n,0))\\ &+\lambda'(t_n,t_{n'})d\left(\boldsymbol{f_{\bar\theta}}(\boldsymbol{f_\theta}(\mathbf{x}+t_n\mathbf{z},t_n,t_{n'}),t_{n'},0),\boldsymbol{f_{\bar\theta}}(\mathbf{x}+t_n\mathbf{z},t_n,0)\right)\Big]\end{aligned} \tag{10}$$

where we set reweighting as $\lambda'(t_n,t_{n'}) = {}^1/|t_n - t_{n'}|$ to keep the loss scale consistent.

**Training from scratch.** We can train BCM from scratch using Equation (10). We empirically found that all the training settings used for iCT (Song & Dhariwal, 2024), including the scheduler function for $N$, the sampling probability for $t_n$ (aka the noise schedule $p(n)$ in (Song et al., 2023; Song & Dhariwal, 2024)), the EMA rate, and more, also works well for BCM. Please refer to Appendix C.1 for more details on these settings and hyperparameters. We summarize the training process in Algorithm 1 and compare the training of CT, CTM, and BCT in Table 2 in Appendix E.

**Bidirectional Consistency Fine-tuning.** To reduce the training cost and increase scalability, we can also initialize BCM with a pretrained consistency model and fine-tune it using Equation (10) to incorporate bidirectional consistency. Recall that BCM takes in three arguments: 1) the sample $\mathbf{x}_t$ at time $t$, 2) the current time step $t$, and 3) the target time step $u$, and outputs the sample at time $u$, i.e., $\mathbf{x}_u$. In contrast, the standard consistency model only takes the first two inputs and can be viewed as a special case of BCM when the target time step $u$ is set to $0$. Therefore, we aim to initialize BCM such that it preserves the performance of the pretrained CM when $u = 0$. To achieve this, we concatenate the embeddings of $t$ and $u$, then pass the result through a linear layer (without bias vector) to reduce the dimensionality by half. The weight matrix is initialized as $[\mathbf{I},\mathbf{0}]$. The embedding layers for $t$ and $u$ are initialized using the pretrained CM's embedding layer, but they are not tied together during fine-tuning. Other layers are initialized identically to those of the CM. It is easy to check that this initialization effectively ignores $u$, preserving the CM's generation performance without fine-tuning. See Appendix C.1 for more details.

| (a) | (b) | (c) | (d) |

Figure 2: Comparison of different strategies of adding fresh noise in zigzag sampling. (a) 1-step generation. (b) Zigzag sampling with manually added fresh noise, where the new noises drastically alter the content. (c) Zigzag sampling with manually added, fixed noise, i.e., we fix the injected fresh noise in each iteration to be the same as the initial one. We can see that the quality significantly deteriorates. (d) Zigzag sampling with BCM. At each iteration, we apply a small amount of noise and let the network amplify it. We can see that the image content is mostly maintained.

### 3.3 SAMPLING

In this section, we describe the sampling schemes of BCM. Similar to CMs and CTM, BCM supports 1-step sampling naturally. However, BCM's capability to navigate both forward and backward along the PF ODE trajectory allows us to design more complicated multi-step sampling strategies to improve the sample quality. We present two schemes and a combined approach that has empirically demonstrated superior performance in the following.

**Ancestral Sampling.** The most straightforward way for multi-step sampling is to remove noise sequentially. Specifically, we first divide the time horizon $[0, T]$ into $N$ sub-intervals with boundaries $0 = t_0 < t_1 < \cdots < t_N = T$. Then, we sample a noise image $\mathbf{x}_T \sim \mathcal{N}(0, T^2 \mathbf{I})$, and sequentially remove noise with the network: $\mathbf{x}_{t_{n-1}} \leftarrow \boldsymbol{f}_{\boldsymbol{\theta}}(\mathbf{x}_{t_n}, t_n, t_{n-1}), \quad n = N, N-1, ..., 1$. We note that the discretization strategy may differ from the one used during the training of BCM. Since this sampling procedure can be viewed as drawing samples from the conditional density $p_{\mathbf{x}_{t_{n-1}} | \mathbf{x}_{t_n}}(\mathbf{x}_{t_{n-1}} | \mathbf{x}_{t_n})$, we dub it as ancestral sampling, and summarize it in Algorithm 2. We can also view 1-step sampling as ancestral sampling, where we only divide the time horizon into a single interval.

**Zigzag Sampling.** Another effective sampling method (Algorithm 1 in Song et al. (2023)) is to iteratively re-add noise after denoising. Similar to ancestral sampling, we also define a sequence of time steps $t_1 < \cdots < t_N = T$. However, different from ancestral sampling where we gradually remove noise, we directly map $\mathbf{x}_T$ to $\mathbf{x}_0$ by $\boldsymbol{f}_{\boldsymbol{\theta}}(\mathbf{x}_T, T, 0)$. We then add a fresh Gaussian noise to $\mathbf{x}_0$, mapping it from time 0 to time $t_{n-1}$, i.e., $\mathbf{x}_{t_{n-1}} = \mathbf{x}_0 + t_{n-1}\boldsymbol{\sigma}$, where $\boldsymbol{\sigma} \sim \mathcal{N}(0, \mathbf{I})$. This process repeats in this zigzag manner until all the designated time steps are covered. The two-step zigzag sampler effectively reduces FID in CMs (Song et al., 2023) and is theoretically supported (Lyu et al., 2023). However, the fresh noise can alter the content of the image after each iteration, which is undesirable, especially considering that our tasks will both include generation and inversion. One may immediately think that setting the injected noise to be the same as the initial random noise can fix this issue. However, we reveal that this significantly damages the quality of the generated images.

Fortunately, our proposed BCM provides a direct solution, leveraging its capability to traverse both forward and backward along the PF ODE. Specifically, rather than manually reintroducing a large amount of fresh noise, we initially apply a small amount and let the network amplify it. In a nutshell, for iteration $n$ ($n = N, N-1, \cdots, 2$), we have

$$\mathbf{x}_0 \leftarrow \boldsymbol{f}_{\boldsymbol{\theta}}(\mathbf{x}_{t_n}, t_n, 0), \quad \mathbf{x}_{\varepsilon_{n-1}} \leftarrow \mathbf{x}_0 + \varepsilon_{n-1}\boldsymbol{\sigma}, \quad \mathbf{x}_{t_{n-1}} \leftarrow \boldsymbol{f}_{\boldsymbol{\theta}}(\mathbf{x}_{\varepsilon_{n-1}}, \varepsilon_{n-1}, t_{n-1}). \quad (11)$$

where $\varepsilon_{n-1}$ is the scale of the small noises we add in $n$-th iteration, and $\boldsymbol{\sigma} \sim \mathcal{N}(0, \mathbf{I})$ is a fresh Gaussian noise. We detail this scheme in Algorithm 3. To verify its effectiveness, in Figure 2, we illustrate some examples to compare the generated images by 1) manually adding fresh noise, 2) manually adding fixed noise, and 3) our proposed sampling process, i.e., adding a small noise and amplifying it with the network. We can clearly see our method maintains the generated content.

**Combination of Both.** Long jumps along the PF ODE in zigzag sampling can lead to accumulative errors, especially at high noise levels, which potentially hampers further improvements in sample quality. Therefore, we propose a combination of ancestral sampling and zigzag sampling. Specifically, we first perform ancestral sampling to rapidly reduce the large initial noise to a more manageable noise scale and then apply zigzag sampling within this small noise level. We describe this combined process in Algorithm 4. We empirically found this combination results in superior sample quality compared to employing either ancestral sampling or zigzag sampling in isolation. We provide ablation to evidence the effectiveness of this combination in Appendix B.2.3.

## 3.4 INVERSION

BCM inverts an image following the same principle of sampling. Specifically, we also set an increasing sequence of noise scales $\varepsilon = t_1 < t_2 < \cdots < t_N \leqslant T$. Note that, in contrast to the generation process, it is not always necessary for $t_N$ to equal $T$. Instead, we can adjust it as a hyperparameter based on the specific tasks for which we employ inversion. Then, given an image $\mathbf{x}_0$, we first inject a small Gaussian noise by $\mathbf{x}_{t_1} = \mathbf{x}_0 + \varepsilon\boldsymbol{\sigma}$, and then sequentially add noise with the network, i.e., $\mathbf{x}_{t_{n+1}} = f(\mathbf{x}_{t_n}, t_n, t_{n+1}), \quad n = 1, 2, \cdots, N-1$. The adoption of small initial noise is due to the observation that the endpoint of the time horizon is less effectively covered and learned during training, as discussed in Appendix B.2.4. Empirically, we find this minor noise does not change the image's content and leads to lower reconstruction errors when $\varepsilon \approx 0.07$. One may also include denoising steps interleaved with noise magnifying steps, like zigzag sampling, but we find it helps little in improving inversion quality. We summarize the inversion procedure in Algorithm 5.

## 4 EXPERIMENTS AND RESULTS

In this section, we first present the results of the two basic functions of BCM, i.e., generation and inversion. Then in Section 4.3, we show some potential applications enabled by these two functions.

## 4.1 IMAGE GENERATION

We first evaluate the image generation performance of BCM. On CIFAR-10, we train our BCM from scratch, while on ImageNet-64, we first train an iCT model, and fine-tune BCM from it, as we discussed in Section 3.2. When reproducing iCT on ImageNet-64, we encountered instability during training, and found it difficult to reproduce the reported FID by Song & Dhariwal (2024). To mitigate the performance gap, we removed the attention layer at the 32x32 resolution and added normalization to the QKV attention, following (Karras et al., 2024). We empirically find these simple modifications bring positive influences on performance. We include the hyperparameters for training and sampling in Appendices C.1 and C.2. We report image quality and NFE for CIFAR-10 and ImageNet-64 in Table 1, and we visualize some generated samples Figures 3 and 4 and appendix H. As we can see,

- both ancestral sampling and zigzag sampling can improve the sample quality, while the combination of both further yields the best performance;

- our proposed BCM achieves comparable or even better results within at least one order of magnitude fewer NFEs compared with diffusion models; BCM presents better results with even fewer NFEs then the methods boosting sampling speed through fast samplers; it is also comparable to modern distillation approaches like EM distillation by Xie et al. (2024);

- comparing with CMs, iCT (Song & Dhariwal, 2024) surpasses BCM for 1-step sampling. This is not surprising, given that BCMs tackle a significantly more complex task compared to CMs. However, as the number of function evaluations (NFEs) increases, following the sampling strategies detailed in Section 3.3, BCM starts to outperform. This indicates that *our approach allows the model to learn bidirectional consistency without hurting the generation performance*;

- our model's performance still falls short of CTM's (Kim et al., 2024). However, CTM relies heavily on adversarial loss and uses LPIPS (Zhang et al., 2018) as the distance measure, which can have feature leakage (Song & Dhariwal, 2024; Kynkäänniemi et al., 2023). ECT (Geng et al., 2024) also outperforms BCM on ImageNet-64 with 2 NFEs. However, we note that their approach—fine-tuning from a pretrained diffusion and using the advanced EDM2 architecture (Karras et al., 2024)—is orthogonal to ours. Our method could be combined with theirs for potential improvements in both performance and training speed.

Another interesting phenomenon we observed on ImageNet-64 is that our fine-tuned BCM actually outperforms its initialization (i.e., our pretrained iCT) on generation. In contrast, we empirically find fine-tuning iCT with the same training budget did not yield any improvement. We suspect this is due to the "soft" trajectory constraint in the bidirectional training objective, which may act as a form of regularization, aiding optimization. We leave further investigation of this for future work.

Table 1: Sample quality on CIFAR-10 (left) and ImageNet-64 (right). We train BCM from scratch on CIFAR-10 and fine-tune it using our reproduced iCT model on ImageNet-64. *Results estimated from Figure 13 in Kim et al. (2024). †For our BCM and BCM-deep, we use ancestral sampling when NFE=2, zigzag sampling when NFE=3, and the combination of both when NFE=4. Our results indicate that ancestral and zigzag sampling can individually improve FID, and their combination can achieve even better performance. **Results by our reproduction.

| METHOD | NFE (↓) | FID (↓) | IS (↑) |
|---|---|---|---|
| **Diffusion with Fast Sampler / Distillation / Fine-tuning** | | | |
| DDIM (Song et al., 2021a) | 10 | 8.23 | - |
| DPM-solver-fast (Lu et al., 2022a) | 10 | 4.70 | - |
| AMED-plugin (Zhou et al., 2023) | 5 | 6.61 | - |
| Progressive Distillation (Salimans & Ho, 2022) | 1 | 8.34 | 8.69 |
| Diff-Instruct (Luo et al., 2024) | 1 | 4.53 | - |
| CD (LPIPS, Song et al., 2023) | 1 | 3.55 | 9.48 |
| | 2 | 2.93 | 9.75 |
| CTM (LPIPS, GAN loss, Kim et al., 2024) | 1 | 1.98 | - |
| | 2 | 1.87 | - |
| CTM (LPIPS, w/o GAN loss, Kim et al., 2024) | 1 | > 5.00* | - |
| ECT (Geng et al., 2024) | 2 | 2.11 | - |
| **Direct Generation** | | | |
| EDM (Karras et al., 2022) | 35 | 2.04 | 9.84 |
| CT (LPIPS, Song et al., 2023) | 1 | 8.70 | 8.49 |
| | 2 | 5.83 | 8.85 |
| CTM (LPIPS, GAN loss, Kim et al., 2024) | 1 | 2.39 | - |
| iCT (Song & Dhariwal, 2024) | 1 | 2.83 | 9.54 |
| | 2 | 2.46 | 9.80 |
| iCT-deep (Song & Dhariwal, 2024) | 1 | 2.51 | 9.76 |
| | 2 | 2.24 | 9.89 |
| **BCM (ours)†** | 1 | 3.10 | 9.45 |
| | 2 | 2.39 | 9.88 |
| | 3 | 2.50 | 9.82 |
| | 4 | 2.29 | 9.92 |
| **BCM-deep (ours)†** | 1 | 2.64 | 9.67 |
| | 2 | 2.36 | 9.86 |
| | 3 | 2.19 | 9.94 |
| | 4 | 2.07 | 10.02 |

| METHOD | NFE (↓) | FID (↓) |
|---|---|---|
| **Diffusion with Fast Sampler / Distillation / Fine-tuning** | | |
| DDIM (Song et al., 2021a) | 10 | 18.3 |
| DPM-solver-fast (Lu et al., 2022a) | 10 | 7.93 |
| AMED-solver (Zhou et al., 2023) | 5 | 10.74 |
| Progressive Distillation (Salimans & Ho, 2022) | 1 | 7.88 |
| Diff-Instruct (Luo et al., 2024) | 1 | 5.57 |
| EM Distillation (Xie et al., 2024) | 1 | 2.20 |
| CD (LPIPS, Song et al., 2023) | 1 | 6.20 |
| | 2 | 4.70 |
| CTM (LPIPS, GAN loss, Kim et al., 2024) | 1 | 1.92 |
| | 2 | 1.73 |
| ECT-XL (Geng et al., 2024) | 2 | 1.67 |
| **Direct Generation** | | |
| EDM2-L/XL (Karras et al., 2024) | 63 | 1.33 |
| CT (LPIPS, Song et al., 2023) | 1 | 13.0 |
| | 2 | 11.1 |
| iCT (Song & Dhariwal, 2024) | 1 | 4.02 (4.60**) |
| | 2 | 3.20 (3.40**) |
| iCT-deep (Song & Dhariwal, 2024) | 1 | 3.25 (3.94**) |
| | 2 | 2.77 (3.14**) |
| **BCM (ours)†** | 1 | 4.18 |
| | 2 | 2.88 |
| | 3 | 2.78 |
| | 4 | 2.68 |
| **BCM-deep (ours)†** | 1 | 3.14 |
| | 2 | 2.45 |
| | 3 | 2.61 |
| | 4 | 2.35 |

(a) 1-step (FID 2.64)

(d) 4-step (FID 2.07)

Figure 3: Samples by BCM-deep on CIFAR-10.

(a) 1-step (FID 3.14)          (d) 4-step (FID 2.35)

Figure 4: Samples by BCM-deep on ImageNet-64.

## 4.2 Inversion and Reconstruction

We evaluate BCM's capability for inversion on CIFAR-10 and ImageNet-64, and report the per-dimension mean squared error (scaled to [0,1]) in Figure 5. We also present ODE-based baselines, including DDIM (Song et al., 2021a) and EDM (Karras et al., 2022) for comparison. The hyperparameters for inversion are tuned on a small subset of the training set and are discussed in Appendix C.2. We can see BCM achieves a lower reconstruction error than ODE-based diffusion models, with significantly fewer NFEs, on both data sets.

We visualize the noise generated by BCM at Figure 16 in Appendix G, from which we can see that BCM Gaussianizes the input image as desired. We also provide examples of the reconstruction in Figure 17 in Appendix G. We observe that using only 1 NFE for inversion sometimes introduces slight mosaic artifacts, possibly because both endpoints of the time horizon are less effectively covered and learned during training. Fortunately, the artifact is well suppressed when using more than 1 NFEs. Also, we find that the inversion process can occasionally alter the image context. However, it

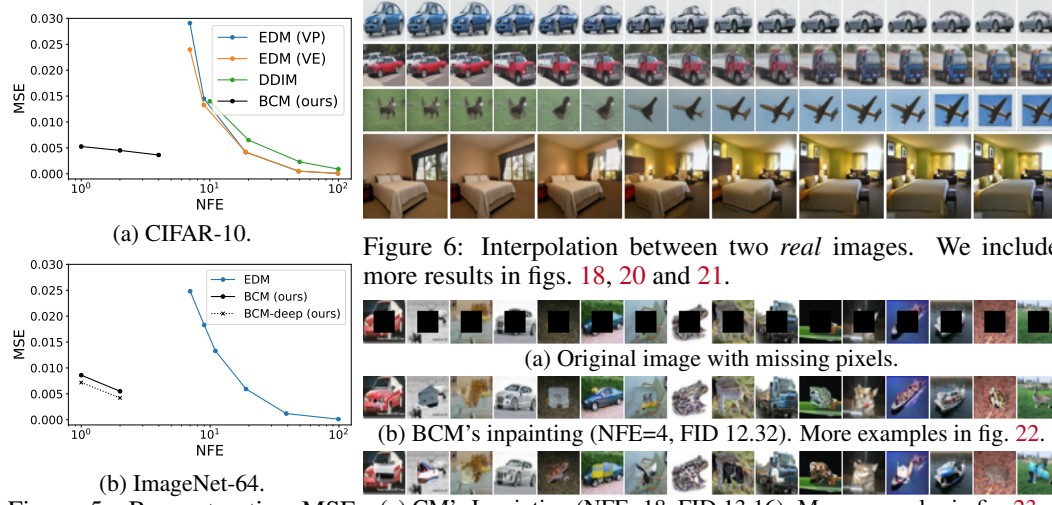

(a) CIFAR-10.

(b) ImageNet-64.

Figure 5: Reconstruction MSE on CIFAR-10 & ImageNet.

Figure 6: Interpolation between two *real* images. We include more results in figs. 18, 20 and 21.

(a) Original image with missing pixels.

(b) BCM's inpainting (NFE=4, FID 12.32). More examples in fig. 22.

(c) CM's Inpainting (NFE=18, FID 13.16). More examples in fig. 23.

Figure 7: Inpainting by BCM and CMs on CIFAR-10 test set.

is important to note that this is not unique to BCM but also occurs in ODE-based diffusion models, such as EDM (see Figures 17f and 17g for examples), even though they employ more NFEs.

### 4.3 APPLICATIONS WITH BIDIRECTIONAL CONSISTENCY

Since BCM inherits the consistency of CMs, it can handle the applications introduced by Song et al. (2023) in the same way, including colorization, super-resolution, stroke-guided image generation, denoising, and interpolation between two *generated* images. However, BCM's unique bidirectional consistency can enhance or enable more applications. In this section, we showcase that BCM can interpolate two *real* images and achieve superior inpainting quality with fewer NFEs. We also demonstrate its downstream applications for blind restoration of compressed images.

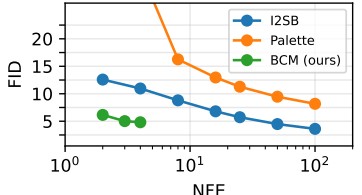

Figure 8: JPEG (QF20) restoration with $I^2SB$, Palette and BCM on ImageNet-64.

**Interpolation.** While CMs can perform interpolations between two *generated* images (Song et al., 2023), our BCM can interpolate between two given *real* images, which is a more meaningful application. Specifically, we can first invert the given images into noises, smoothly interpolate between them, and then map the noises back to images. We illustrate some examples in Figure 6. Here, we note a caveat in the implementation of BCM's interpolation: recall that when inverting an image, we first inject a small initial noise, as described in Section 3.4. In the context of interpolation, we find it crucial to inject *different* initial noises for each of the two given images to avoid sub-optimal results, as illustrated in Figure 19. We defer further discussions to Appendix C.3.1.

**Inpainting.** BCM's bidirectional consistency can also help with inpainting: having an image with some pixels missing, we first add a small noise to the missing pixels and then invert the image by multi-step inversion. Different from our standard inversion algorithm in Algorithm 5, at each iteration, we manually replace the region of missing pixels with a new Gaussian noise, whose scale corresponds to the current time step. In this way, we gradually fill in the missing region to in-distribution noise. In the end, we map the entire noisy image back to time 0, finishing the inpainting. We summarize this process in Algorithm 6, and include more details on the hyperparameters in Appendix C.3.2. We illustrate our BCM inpainting results on CIFAR-10 in Figure 7, with the CM's inpainting results using the algorithms proposed by Song et al. (2023) (Algorithm 4 in (Song et al., 2023)) for comparison. Our method produces better inpainting results within much fewer NFEs.

**Blind restoration of compressed image.** BCM's bidirectional consistency can offer broader applications. For example, we can invert an image with OOD artifacts to the noise space and re-generate

the image from the noise. During inversion, the model maps OOD regions to in-distribution noise, effectively eliminating these artifacts. One potential application leveraging this property is blind restoration of compressed image, where OOD artifacts originate from compression. We prove this concept by applying BCM to restore images compressed by JPEG. We present the results in Figure 8, and also include two established restoration approaches, I$^2$SB (Liu et al., 2023) and Palette (Saharia et al., 2022), for comparison. We include hyperparameters for our approach and these baselines in Appendix C.3.3. Surprisingly, BCM achieves better performance compared to both I$^2$SB and Palette when NFE is limited. Note that I$^2$SB and Palette require paired data under known degradation for training, while BCM is fully zero-shot and blind. This result strongly evidences the potential of BCM in downstream inversion tasks.

## 5 RELATED WORKS

**Accelerating Diffusion's Generation.** In addition to CM and CTM we discussed in Section 2, many attempts have also been made to accelerate the generation of DMs. Existing methods include faster ODE solvers (Song et al., 2021a; Zhang & Chen, 2022; Lu et al., 2022b; Zhou et al., 2023) and distillation (Salimans & Ho, 2022; Luhman & Luhman, 2021; Zheng et al., 2023; Luo et al., 2024; Xie et al., 2024). However, these ODE solvers generally require 5-10 steps for satisfactory samples, and the distillation approach may cap the performance at the level of the pretrained diffusion model, as highlighted by Song & Dhariwal (2024). On the other hand, while distillation approaches have achieved impressive generation quality, an intrinsic property of CMs is their ability to enable fast traversal along ODEs. This allows for the design of broader applications; for example, Zhang et al. (2024) leverage this property to develop an efficient importance sampler. Our proposed BCM further enriches this by enabling both forward and backward traversal.

**Inversion.** Inversion of an input image is a key step in downstream tasks like image editing (Mokady et al., 2023; Wallace et al., 2023; Hertz et al., 2023; Huberman-Spiegelglas et al., 2023). Current methods typically include encoders (Richardson et al., 2021; Tov et al., 2021), optimization (Mokady et al., 2023; Abdal et al., 2019), and model fine-tuning or modulation (Roich et al., 2022; Alaluf et al., 2022). For diffusion-based models, the most common inversion method is based on solving PF ODE backward, as first proposed by Song et al. (2021a). However, since the ODE solver relies on local linearization, this ODE-based inversion typically provides an insufficient reconstruction, especially when using fewer diffusion steps. One way to address this issue is to perform gradient optimization using the ODE-based inversion as an initialization, as introduced by Mokady et al. (2023). Our proposed BCM can provide better inversion with fewer NFEs and is compatible with Mokady et al. (2023)'s approach for a better reconstruction quality.

## 6 CONCLUSIONS AND LIMITATIONS

In this work, we introduce the Bidirectional Consistency Model (BCM), enhancing upon existing consistency models (Song et al., 2023; Song & Dhariwal, 2024; Kim et al., 2024) by establishing a stronger consistency. This consistency ensures that points along the same trajectory of the probability flow (PF) ODE map to each other, thereby unifying generation and inversion tasks within one framework. By exploiting its bidirectional consistency, we devise new sampling schemes and showcase applications in a variety of downstream tasks. Unlike other distillation approaches, a notable property of CMs is their ability to enable fast traversal along the entire PF ODE. Our proposed BCM further enriches this, and we believe that it will open a new avenue for further exploration.

One limitation of our method is that while employing more steps in generation or inversion can initially enhance results, the performance improvements tend to plateau quickly. Increasing the Number of Function Evaluations (NFEs) beyond a certain point does not yield further performance gains. This is similar to CMs, where there is no performance gain with more than 2 NFEs. A potential solution involves employing the parameterization and tricks proposed by Kim et al. (2024). Additionally, our method delivers imperfect inversion, which sometimes alters the image content. However, we should note that this also happens in ODE-based DMs. Future work can involve developing more accurate inversion techniques, like the approach by Wallace et al. (2023).

## BROADER IMPACT AND ETHICS STATEMENTS

BCM can accelerate the generation and inversion of diffusion models, which may pose a risk of generating or editing images with harmful or inappropriate content, such as deepfake images or offensive material. Strong content filtering or even regulatory rules are required to prevent the creation of unethical or harmful content.

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

# BIDIRECTIONAL CONSISTENCY MODELS: APPENDIX

## A    ALGORITHMS

---

**Algorithm 2** BCM's ancestral sampling

---

**Input:** Network $\boldsymbol{f_\theta}(\cdot, \cdot, \cdot)$, time steps $0 = t_0 < t_1 < \cdots < t_N = T$, initial noise $\mathbf{x}_T$.
**Output:** Generated image $\mathbf{x}_{t_0}$.
   $\mathbf{x}_{t_N} \leftarrow \mathbf{x}_T$.
   **for** $n = N, \cdots, 1$ **do**
      $\mathbf{x}_{t_{n-1}} \leftarrow \boldsymbol{f_\theta}(\mathbf{x}_{t_n}, t_n, t_{n-1})$.        ▷ Denoise image from time step $t_n$ to $t_{n-1}$.
   **end for**
   **Return:** $\mathbf{x}_{t_0}$.

---

**Algorithm 3** BCM's zigzag sampling

---

**Input:** Network $\boldsymbol{f_\theta}(\cdot, \cdot, \cdot)$, time steps $t_1 < \cdots < t_N = T$, manually-added noise scale at each time step $\varepsilon_1, \ldots, \varepsilon_{N-1}$, initial noise $\mathbf{x}_T$.
**Output:** Generated image $\mathbf{x}$.
   $\mathbf{x}_{t_N} \leftarrow \mathbf{x}_T$.
   **for** $n = N, \cdots, 2$ **do**
      $\mathbf{x} \leftarrow \boldsymbol{f_\theta}(\mathbf{x}_{t_n}, t_n, 0)$.        ▷ Denoise image from time step $t_n$ to 0.
      $\boldsymbol{\sigma} \sim \mathcal{N}(0, \boldsymbol{I})$, and $\mathbf{x}_{\varepsilon_{n-1}} \leftarrow \mathbf{x} + \varepsilon_{n-1}\boldsymbol{\sigma}$.        ▷ Add small fresh noise.
      $\mathbf{x}_{t_{n-1}} \leftarrow \boldsymbol{f_\theta}(\mathbf{x}_{\varepsilon_{n-1}}, \varepsilon_{n-1}, t_{n-1})$.        ▷ Amplify noise by network.
   **end for**
   $\mathbf{x} \leftarrow \boldsymbol{f_\theta}(\mathbf{x}_{t_1}, t_1, 0)$.
   **Return:** $\mathbf{x}$.

---

**Algorithm 4** Combination of ancestral and zigzag sampling

---

**Input:** Network $\boldsymbol{f_\theta}(\cdot, \cdot, \cdot)$, ancestral time steps $t_1 < \cdots < t_N = T$, zigzag time steps $\tau_1 < \cdots < \tau_M = t_1$, manually-added noise scale at each time step $\varepsilon_1, \ldots, \varepsilon_{M-1}$, initial noise $\mathbf{x}_T$.
**Output:** Generated image $\mathbf{x}$.
   $\mathbf{x}_{t_N} \leftarrow \mathbf{x}_T$.
   # Ancestral sampling steps
   **for** $n = N, \cdots, 2$ **do**
      $\mathbf{x}_{t_{n-1}} \leftarrow \boldsymbol{f_\theta}(\mathbf{x}_{t_n}, t_n, t_{n-1})$.        ▷ Denoise image from time step $t_n$ to $t_{n-1}$.
   **end for**
   # Zigzag sampling steps
   $\mathbf{x}_{\tau_M} \leftarrow \mathbf{x}_{t_1}$.
   **for** $m = M, \cdots, 2$ **do**
      $\mathbf{x} \leftarrow \boldsymbol{f_\theta}(\mathbf{x}_{\tau_m}, \tau_m, 0)$.        ▷ Denoise image from time step $\tau_m$ to 0.
      $\boldsymbol{\sigma} \sim \mathcal{N}(0, \boldsymbol{I})$, and $\mathbf{x}_{\varepsilon_{m-1}} \leftarrow \mathbf{x} + \varepsilon_{m-1}\boldsymbol{\sigma}$.        ▷ Add small fresh noise.
      $\mathbf{x}_{\tau_{m-1}} \leftarrow \boldsymbol{f_\theta}(\mathbf{x}_{\varepsilon_{m-1}}, \varepsilon_{m-1}, \tau_{m-1})$.        ▷ Amplify noise by network.
   **end for**
   $\mathbf{x} \leftarrow \boldsymbol{f_\theta}(\mathbf{x}_{\tau_1}, \tau_1, 0)$.
   **Return:** $\mathbf{x}$.

---

---

**Algorithm 5** BCM's inversion

---

**Input:** Network $\boldsymbol{f_\theta}(\cdot, \cdot, \cdot)$, time steps $\varepsilon = t_1 < \cdots < t_N \leqslant T$, initial image $\mathbf{x}_0$.
**Output:** Noise $\mathbf{x}_{t_N}$.
  $\boldsymbol{\sigma} \sim \mathcal{N}(0, \boldsymbol{I}), \quad \mathbf{x}_{t_1} \leftarrow \mathbf{x} + \varepsilon\boldsymbol{\sigma}$.
  **for** $n = 2, \cdots, N$ **do**
    $\mathbf{x}_{t_n} \leftarrow \boldsymbol{f_\theta}(\mathbf{x}_{t_{n-1}}, t_{n-1}, t_n)$.            $\triangleright$ Add noise to image from time step $t_{n-1}$ to $t_n$.
  **end for**
  **Return:** $\mathbf{x}_{t_N}$.

---

**Algorithm 6** BCM's inpainting

---

**Input:** Network $\boldsymbol{f_\theta}(\cdot, \cdot, \cdot)$, time steps $\varepsilon = t_1 < \cdots < t_N \leqslant T$, initial image $\mathbf{x}$, binary image mask $\boldsymbol{\Omega}$ where 1 indicates the missing pixels, initial noise scale for masked region $s$.
**Output:** Inpainted image $\hat{\mathbf{x}}$.
  # Dataset preparation
  $\tilde{\mathbf{x}} \leftarrow \mathbf{x} \odot (1 - \boldsymbol{\Omega}) + \mathbf{0} \odot \boldsymbol{\Omega}$.            $\triangleright$ Create image with missing pixels.
  # Initialization
  $\boldsymbol{\sigma} \sim \mathcal{N}(0, \boldsymbol{I})$.
  $\tilde{\mathbf{x}} \leftarrow \tilde{\mathbf{x}} \odot (1 - \boldsymbol{\Omega}) + s\boldsymbol{\sigma} \odot \boldsymbol{\Omega}$.       $\triangleright$ Manually add small noises to missing pixels.
  # Inversion steps
  $\boldsymbol{\sigma}' \sim \mathcal{N}(0, \boldsymbol{I})$.
  $\mathbf{x}_{t_1} \leftarrow \tilde{\mathbf{x}} + \varepsilon\boldsymbol{\sigma}'$.       $\triangleright$ Manually add initial noise for inversion (similar to Algorithm 5)
  **for** $n = 2, \cdots, N$ **do**
    $\mathbf{x}_{t_n} \leftarrow \boldsymbol{f_\theta}(\mathbf{x}_{t_{n-1}}, t_{n-1}, t_n)$.          $\triangleright$ Add noise to image from time step $t_{n-1}$ to $t_n$.
    $\boldsymbol{\sigma}'' \sim \mathcal{N}(0, \boldsymbol{I})$.
    $\mathbf{x}_{t_n} \leftarrow \mathbf{x}_{t_n} \odot (1 - \boldsymbol{\Omega}) + t_n\boldsymbol{\sigma}'' \odot \boldsymbol{\Omega}$.    $\triangleright$ Replace missing region with in-distribution noise.
  **end for**
  # Generation steps
  (Here, we exemplify with 1-step sampling, but note that multi-step schemes can also be used)
  $\mathbf{x}_0 \leftarrow \boldsymbol{f_\theta}(\mathbf{x}_{t_N}, t_N, 0)$.
  $\hat{\mathbf{x}} \leftarrow \tilde{\mathbf{x}} \odot (1 - \boldsymbol{\Omega}) + \mathbf{x}_0 \odot \boldsymbol{\Omega}$       $\triangleright$ Leave the region which is not missing unchanged.
  **Return:** $\hat{\mathbf{x}}$.

---

**Algorithm 7** BCM's inpainting with refinement

---

**Input:** Network $\boldsymbol{f_\theta}(\cdot, \cdot, \cdot)$, BCM inpainting time steps $\varepsilon = t_1 < \cdots < t_N \leqslant T$, Refinement time steps $\tau_1 > \cdots > \tau_M$, initial image $\mathbf{x}$, binary image mask $\boldsymbol{\Omega}$ where 1 indicates the missing pixels, initial noise scale for masked region $s$.
**Output:** Inpainted image $\hat{\mathbf{x}}$.
  # Initialize with BCM's inpainting
  $\hat{\mathbf{x}} \leftarrow$ Algorithm 6$(\boldsymbol{f_\theta}, t_1, \ldots, t_N, \mathbf{x}, \boldsymbol{\Omega}, s)$.
  # Refine
  **for** $m = 1, \cdots, M$ **do**
    $\hat{\mathbf{x}} \sim \mathcal{N}(\hat{\mathbf{x}}, \tau_m^2 \boldsymbol{I})$.
    $\hat{\mathbf{x}} \leftarrow \boldsymbol{f_\theta}(\hat{\mathbf{x}}, \tau_m, 0)$.
    $\hat{\mathbf{x}} \leftarrow \mathbf{x} \odot (1 - \boldsymbol{\Omega}) + \hat{\mathbf{x}} \odot \boldsymbol{\Omega}$.
  **end for**
  **Return:** $\hat{\mathbf{x}}$.

---

# B    ADDITIONAL EXPERIMENTS

## B.1    ADDITIONAL APPLICATIONS WITH BIDIRECTIONAL CONSISTENCY

Here, we provide more demonstrations of BCM's application.

### B.1.1    INPAINTING ON HIGHER-RESOLUTION IMAGES

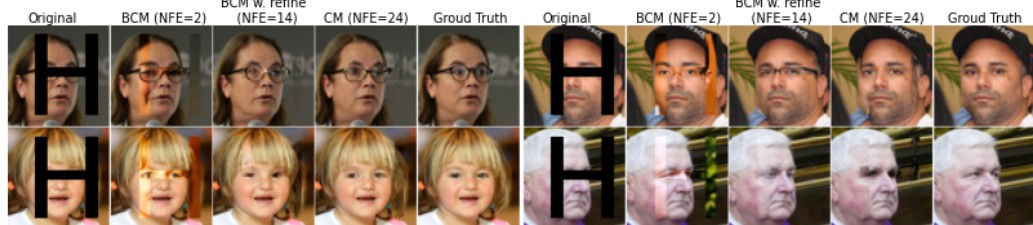

(a) FFHQ $64 \times 64$

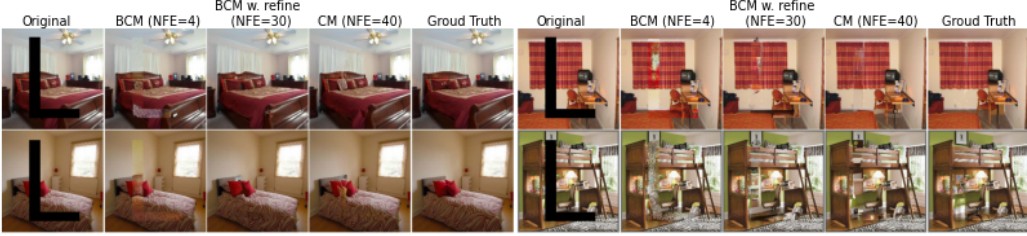

(b) LSUN $256 \times 256$

Figure 9: Inpainting on FFHQ and LSUN Bedroom. We compare inpainting with BCM (Algorithm 6 and Algorithm 7) with CM's inpainting. We visualize more uncurated examples in figs. 24 and 25.

In the main text, we demonstrate that BCM can yield better inpainting results with fewer NFEs on CIFAR-10 images compared with CM's inpainting algorithm proposed by Song et al. (2023). On high-resolution images, however, we find that simply following the same algorithm as CIFAR-10 (Algorithm 6) is suboptimal, possibly because the correlation between pixels is smaller than that in low-resolution. Fortunately, we find Algorithm 6 can quickly fill the missing region with imperfect but sensible results. Therefore, we can add a few more steps to refine the inpainting results by iteratively adding some small noise to the inpainting regions and denoising by BCM. We summarize this algorithm in Algorithm 7. This can be viewed as a combination of BCM's inpainting by Algorithm 6 with CM's inpainting algorithm proposed by Song et al. (2023). We observe that this combination can reduce the NFEs without hurting the inpainting performance.

We evaluate Algorithms 6 and 7 on FFHQ $64 \times 64$ and LSUN Bedroom $256 \times 256$ in Figure 9 and also compare the results of CM's inpainting. The hyperparameters are included in Appendix C.3.2. As we can see: 1) simply adopting BCM's inpainting as described in Algorithm 6 can fill in relatively sensible content but is imperfect in the overall color and details; 2) CM's inpainting algorithm yields satisfactory performance with significantly more NFEs; and 3) applying BCM's inpainting with refinement as described in Algorithm 7 can achieve a comparable performance with fewer NFEs.

## B.2    ADDITIONAL ABLATION STUDIES AND CONCLUSIONS

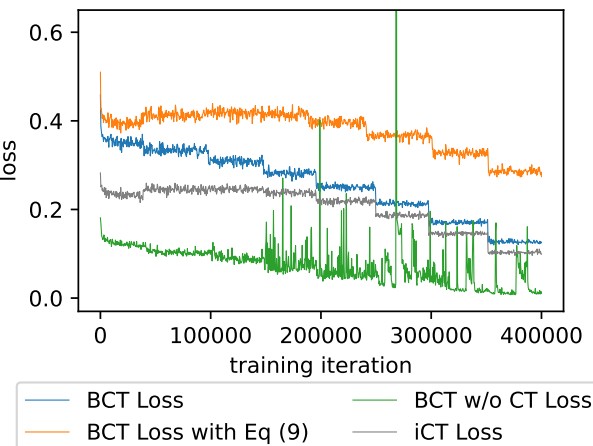

Figure 10: Tracking the loss with different objection functions. We include the loss curve of iCT (Song & Dhariwal, 2024) for reference. We can see that the model with BCT loss defined in Equation (10) converges well. Conversely, the model applying Equation (8) instead of Equation (9) for the soft constraint has a much higher loss at the end of the optimization. While the one without CT loss totally diverges.

### B.2.1    COMPARISON BETWEEN LOSS DEFINED WITH EQUATION (8) AND EQUATION (9)

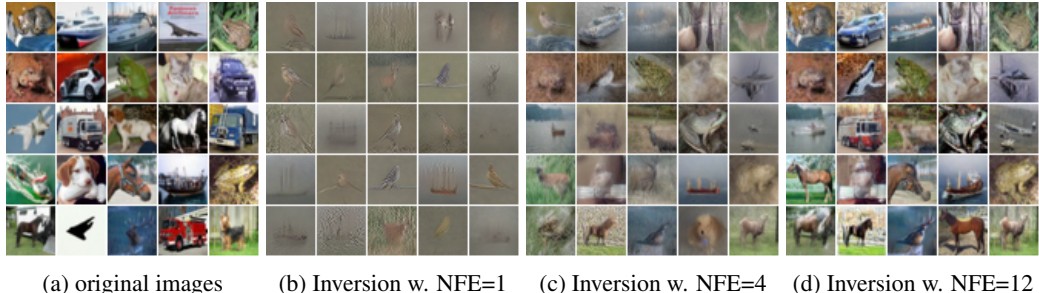

(a) original images       (b) Inversion w. NFE=1       (c) Inversion w. NFE=4       (d) Inversion w. NFE=12

Figure 11: Inversion and reconstruction by BCM trained with Equation (8). We can see the model trained fails to provide an accurate inversion. Even though the images start to look plausible with NFE=12, the content compared with the original images has been significantly changed.

In Section 3.2, we discussed that we optimize Equation (9) instead of Equation (8). Here we provide experimental evidence for our design choice.

We track the loss by models trained with both choices in Figure 10, where we can see that the model trained with Equation (8) features a much higher loss in the end. This echoes its failure in the inversion process: as shown in Figure 11, the model trained with Equation (8) fails to provide an accurate inversion. This is because Equation (8) contains two trajectories, starting from $\mathbf{x}_u$ and $\mathbf{x}_t$. While both of them are along the SDE trajectories starting from the same $\mathbf{x}_0$, they do not necessarily reside on the same PF ODE trajectory; in fact, the probability that they are on the same PF ODE trajectory is 0. On the contrary, Equation (9) bypasses this issue since it only involves trajectories starting from the same $\mathbf{x}_t$.

### B.2.2    ABLATION OF CT LOSS

Recall our final loss function has two terms, the soft trajectory constraint term and the CT loss term. We note that the soft constraint defined in Equation (9) can, in principle, cover the entire trajectory,

so it should also be able to learn the mapping from any time step $t$ to 0, which is the aim of CT loss. However, we find it crucial to include CT loss in our objective. We provide the loss curve trained without CT loss term in Figure 10, where we can see the training fails to converge. For further verification, we also visualize the images generated by the model trained with full BCM loss and the model trained without CT loss term after 200k iterations in Figure 12. We can clearly see that the model without CT loss cannot deliver meaningful outcomes.

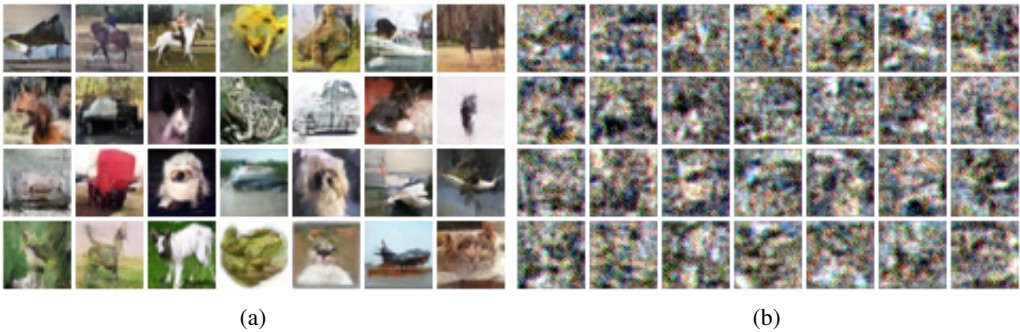

(a)                                                                                 (b)

Figure 12: Images generated by (a) the model trained with full BCM loss for 200k iterations, and (b) the model trained without CT loss term for 200k iterations.

### B.2.3 ALBATION OF DIFFERENT SAMPLING SCHEMES

In this section, we compare image generation quality using different sampling schemes we proposed in Section 3.3. Specifically, we compare FID v.s. NFE on CIFAR-10 using ancestral sampling, zigzag sampling and their combination in Figure 13. As we can see, both zigzag sampling and ancestral sampling offer performance gain over single-step generation. However, as NFE increases, zigzag sampling and ancestral sampling cannot provide further improvement in generation quality. This phenomenon has also been discussed by Kim et al. (2024). However, the combination of both can yield better FID than either of them in isolation.

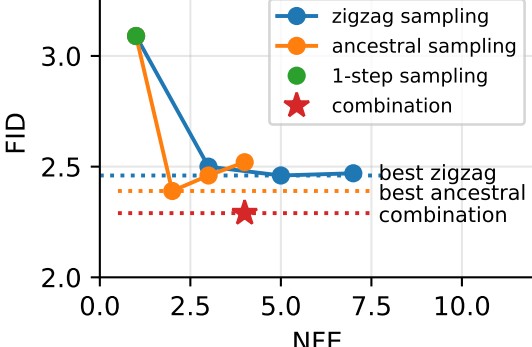

Figure 13: Comparing zigzag sampling, ancestral sampling, and their combination on CIFAR-10. Both zigzag sampling and ancestral sampling offer performance gain over single-step generation, where their combination is better than either in isolation.

### B.2.4 Coverage of Trajectory during Training

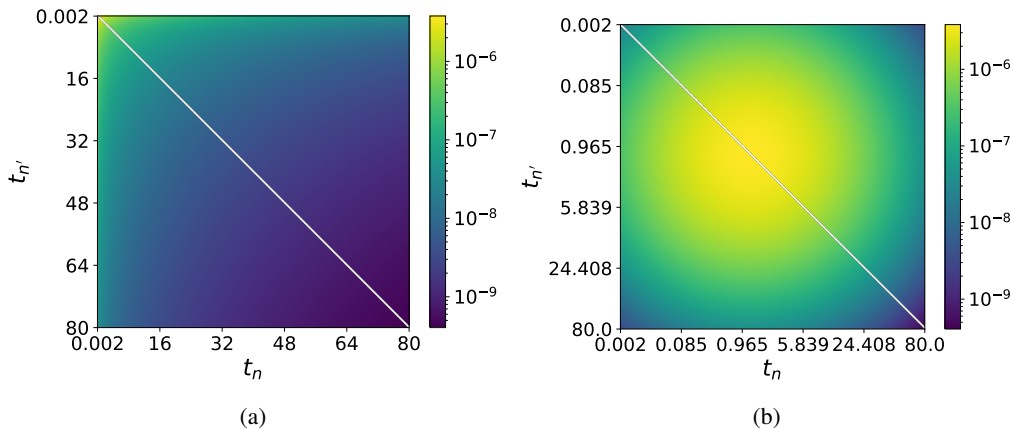

(a)              (b)

Figure 14: Probability mass of $(t_n, t_{n'})$ pair being selected during BCT. We transfer the time axes to log space in (b) for a clearer visualization.

In Figure 14, we visualize the probability of selecting a $(t_n, t_{n'})$ pair during the BCT process, where $(t_n, t_{n'})$ is defined in Equation (10). This offers insights into how well the entire trajectory is covered and trained. We can see that most of the probability mass is concentrated in small-time regions. This reveals some of our observations in the experiments:

- first, in the generation process, we find the combination of a zigzag and ancestral sampling yields optimal performance because the ancestral sampler can rapidly jump over the large-time regions, which we cover less in training;

- second, it also explains why adding a small initial noise in inversion helps: simply adding a small $\varepsilon \approx 0.085$ noise increases the probability of being selected during BCT by more than a thousandfold;

- third, it offers insights into the necessity of incorporating CT loss into our final objective, as defined in Equation (10): while theoretically, the soft constraint is expected to cover the entire trajectory, including boundary conditions, it is highly inefficient in practice. Therefore, explicitly including CT loss to learn the mapping from any noise scale $t$ to $0$ is crucial.

This also points out some future directions to improve BCM. For example, we can design a better sampling strategy during training to ensure a better coverage of the entire trajectory. Or using different sampling strategy for CT loss term and the soft trajectory constraint term.

## C Experiment Details

In this section, we provide the experiment details omitted from the main paper.

### C.1 Training Settings

**CIFAR-10.** For all the experiments on CIFAR-10, following the optimal settings in Song & Dhariwal (2024), we use a batch size of 1,024 with the student EMA decay rate of 0.99993, scale parameter in Fourier embedding layers of 0.02, and dropout rate of 0.3 for 400,000 iterations with RAdam optimizer Liu et al. (2020) using learning rate 0.0001. We use the NCSN++ network architecture proposed by Song et al. (2021b), with the modification described in Section 3.1. In our network parameterization, we set $\sigma_{\text{data}} = 0.5$ following Karras et al. (2022) and Song et al. (2023). Regarding other training settings, including the scheduler function for $N(\cdot)$, the sampling probability for $t_n$ (aka the noise schedule $p(n)$ in (Song et al., 2023; Song & Dhariwal, 2024)), the distance measure $d(\cdot, \cdot)$, we follow exactly Song & Dhariwal (2024), and restate below for completeness's

sake:

$$d(\mathbf{x}, \mathbf{y}) = \sqrt{||x - y||^2 + c^2} - c, \quad c = 0.00054\sqrt{d}, \quad d \text{ is data dimensionality,} \tag{12}$$

$$p(n) \propto \text{erf}\left(\frac{\log(t_{n+1} - P_{\text{mean}})}{\sqrt{2}P_{\text{std}}}\right) - \text{erf}\left(\frac{\log(t_n - P_{\text{mean}})}{\sqrt{2}P_{\text{std}}}\right), \quad P_{\text{mean}} = -1.1, \quad P_{\text{std}} = 2.0, \tag{13}$$

$$N(k) = \min(s_0 2^{\lfloor k/K' \rfloor}, s_1) + 1, \quad s_0 = 10, \quad s_1 = 1280, \quad K' = \left\lfloor \frac{K}{\log_2\lceil s_1/s_0 \rceil + 1} \right\rfloor, \tag{14}$$

$$K \text{ is the total training iterations}$$

$$t_n = \left(t_{\text{min}}^{1/\rho} + \frac{n-1}{N(k)-1}\left(t_{\text{max}}^{1/\rho} - t_{\text{min}}^{1/\rho}\right)\right)^\rho, \quad t_{\text{min}} = 0.002, \quad t_{\text{max}} = 80, \quad \rho = 7. \tag{15}$$

**FFHQ** $64 \times 64$ **and LSUN Bedroom** $256 \times 256$. Except the differences stated below, we train BCMs on FFHQ $64 \times 64$ and LSUN Bedroom $256 \times 256$ using exactly the same settings:

- For FFHQ $64 \times 64$, we use a batch size of 512 but still train the model for 400,000 iterations.
- For LSUN Bedroom $256 \times 256$, we (i) use a dropout rate of 0.2, (ii) adopt the network architecture configuration on this dataset in Song et al. (2021b), (iii) use a batch size of 128, and (iv) adopt a different discretization curriculum from Equation (14). Specifically, in Equation (14), the training procedure is divided into 8 stages. Each stage uses a different $N$ and has $K/8$ iterations. In our experiments, we still divide the entire training procedure into 8 stages, and still set $N(k) = \min(s_0 2^{j-1}, s_1) + 1$ at stage $j$, for $j = 1, 2, \cdots, 8$. However, we do not divide these stages evenly. Instead, we assign 75,000, 60,000, 45,000, 30,000 iterations to the 1st/2nd stage, 3rd/4th stage, 5th/6th stage, 7th/8th stage, respectively [3]. The total number of iterations is 420,000.

Moreover, since our experiments include many applications that are necessarily conducted on test sets (e.g., inpainting), we split FFHQ into a training set of 69,000 images and a test set containing the last 1,000 images. For LSUN Bedroom, we use its official validation set as test set. We implement our model and training algorithm based on the codes released by Song et al. (2023) at `https://github.com/openai/consistency_models_cifar10` (Apache-2.0 License).

We specially emphasize that the settings we use to train FFHQ $64 \times 64$ and LSUN Bedroom $256 \times 256$ are not fine-tuned to optimal (in fact, most of them have not been tuned at all). For instance, Song et al. (2023) trained LSUN Bedroom $256 \times 256$ with a significantly larger batch size of 2,048 for 1,000,000 iterations, meaning that their model has seen more than $38\times$ samples than ours during training. Therefore, we note that our settings for these two datasets should NOT be used as guidance for training the models to optimal performance, and our results are only for demonstration purposes and should NOT be directly compared with other baselines unless under aligned settings.

**ImageNet-64.** Instead of training from scratch, we apply bidirectional consistency fine-tuning on ImageNet-64. Specifically, we first train an iCT on ImageNet-64, and then initialize and fine-tune BCM from it. Since there is no official implementation or available checkpoints for iCT, we reproduce it by ourselves. Therefore, we first state our training settings for iCT:

**Reproducing iCT on ImageNet-64.** Our training setting follows Song & Dhariwal (2024) closely. Most of the training parameters for ImageNet-64 are consistent with those used for training BCM on CIFAR-10, with the exceptions noted below:

- We use a batch size of 4096 and train the model for 800,000 iterations.
- The EMA decay rate is set to 0.99997.
- We adopt the ADM architecture and remove AdaGN following Song & Dhariwal (2024).
- Instead of Fourier Embeddings, we employ the default positional embedding.
- The dropout rate is set to 0.2 and only applied to convolutional layers whose feature map resolution is smaller or equal to $16 \times 16$.
- We use mix-precision training.

---

[3] We assign more iterations to earlier stages with small $N$, as we empirically find that it yields better performance *given limited computing budget*.

Additionally, when reproducing iCT on ImageNet-64, following EDM2 (Karras et al., 2024), we slightly modify the network architecture as follows:

- The self-attention layer at resolution $32 \times 32$ is removed.
- The $Q, K$ and $V$ vectors are normalized in the self-attention layers.

We empirically find these simple modifications bring positive influences on performance.

**Fine-tuning BCM on ImageNet-64.** We initialize our BCM with the pretrained iCT weights and fine-tune it using Equation (10). Recall that BCM takes in three arguments: 1) the sample $\mathbf{x}_t$ at time $t$, 2) the current time step $t$, and 3) the target time step $u$, and outputs the sample at time $u$, i.e., $\mathbf{x}_u$. In contrast, the pretrained iCT model only takes the first two inputs and can be viewed as a special case of BCM when the target time step $u$ is set to 0. Therefore, we aim to initialize BCM such that it preserves the performance of the pretrained CM when $u = 0$. To achieve this, we carefully initialize the BCM as follows:

- In iCT, the time embedding of $t$ is passed through a 2-layer MLP with SiLU activation, after which it is fed into the residual blocks. At the initialization of BCM, we duplicate the 2-layer MLP (with the same pretrained weights) for $u$. However, during fine-tuning, the MLPs for $t$ and $u$ are not tied.
- We do not want to expand the input dimension of the pretrained residual blocks, so first concatenate the embedding of $t$ and $u$ together, and then reduce its dimension by half with a simple linear layer (without bias). No additional activation function is applied for this simple linear layer.
- In order to maintain the generation quality of the pretrained iCT, we initialize this linear projection as $[\mathbf{I}, \mathbf{0}]$, in which $\mathbf{I}$ represents an identity matrix and $\mathbf{0}$ represents a zero matrix. As a result, the influence brought by the newly introduced time embedding of $u$ is zeroed at the beginning and learned gradually during fine-tuning.

With the modified architecture initialized with the pretrained CM weights, we tune the models using our proposed BCT loss till convergence, using the same learning rate, ema rate, etc., as in iCT. The only difference is that we start the fine-tuning from $N = 320$ and increase it to $N = 480$ and subsequently $N = 640$. Specifically,

- we tune BCM for 210k, 16k and 8k iterations respectively at $N = 320, 480$ and 640.
- we tune BCM-deep for 360k, 100k and 10k iterations respectively at $N = 320, 480$ and 640.

The number of iterations for different values of $N$ is determined empirically: we increase $N$ when we observe no performance improvements by extending the training duration at the same $N$.

## C.2 Sampling and Inversion Configurations

Here we provide hyperparameters for all the experiments on CIFAR-10 and ImageNet-64 $\times$ 64 to reproduce the results in the main paper.

**Sampling.** On *CIFAR-10*: for BCM (not deep), we use ancestral sampling with $t_1 = 1.2$ for NFE$= 2$, zigzag sampling with $\varepsilon_1 = 0.2, t_1 = 0.8$ for NFE$= 3$ and the combination of ancestral sampling and zigzag sampling with $t_1 = 1.2, \varepsilon_1 = 0.1, \tau_1 = 0.3$ for NFE$= 4$. For BCM-deep, we use ancestral sampling with $t_1 = 0.7$ for NFE$= 2$, zigzag sampling with $\varepsilon_1 = 0.4, t_1 = 0.8$ for NFE$= 3$ and the combination with $t_1 = 0.6, \varepsilon_1 = 0.14, \tau_1 = 0.3$ for NFE$= 4$. On *ImageNet-64 $\times$ 64*: for both conditional BCM and BCM-deep, we use ancestral sampling with $t_1 = 2.4$ for NFE$= 2$, zigzag sampling with $\varepsilon_1 = 0.1, t_1 = 1.2$ for NFE$= 3$ and the combination of ancestral sampling and zigzag sampling with $t_1 = 3.0, \varepsilon_1 = 0.12, \tau_1 = 0.4$ for NFE$= 4$.

**Inversion.** On *CIFAR-10*: we set $\varepsilon = t_1 = 0.07, t_2 = 6.0$ and $t_3 = T = 80.0$ for NFE$= 2$ and $t_2 = 1.5, t_3 = 4.0, t_4 = 10.0, t_5 = T = 80.0$ for NFE$= 4$. These hyperparameters are tuned on 2,000 training samples, and we find them generalize well to all test images. On *ImageNet-64 $\times$ 64*: for both BCM and BCM-deep, we set $\varepsilon = t_1 = 0.07, t_2 = 15.0$ and $t_3 = T = 80.0$ for NFE$= 2$. In

the experiments we always use 1-step generation to map the inverted noise to reconstructed images (though using more than one step may improve the results) and evaluate the per-dimension MSE between the original images and their reconstructed counterparts.

We highlight that the hyperparameters are relatively robust for 2-step sampling/inversion, and the trend that the combination of ancestral and zigzag sampling is superior is also general. However, to achieve optimal performance with more steps, the tuning of each specific time step may require great effort. We should note that similar effort is also required in CMs, where Song et al. (2023) use ternary search to optimize the time steps.

**Inversion Baselines.** For DDIM, we use the reported MSE by Song et al. (2021a); for EDM, we load the checkpoint provided in the official implementation at `https://github.com/NVlabs/edm?tab=readme-ov-file` (CC BY-NC-SA 4.0 License), and re-implement a deterministic ODE solver following Algorithm 1 in (Karras et al., 2022).

### C.3 APPLICATIONS WITH BIDIRECTIONAL CONSISTENCY

Here we provide more details about the applications we demonstrated in Section 4.3 and Appendix B.1.

#### C.3.1 INTERPOLATION

We first invert the two given images $\mathbf{x}_1$ and $\mathbf{x}_2$ to noise at $T = 80.0$ using Algorithm 5. To avoid subscript overloading, in this section, we denote their noise as $\mathbf{z}_1$ and $\mathbf{z}_2$, respectively. Specifically, we find that adopting a 3-step inversion with $\varepsilon = t_1 = 0.07, t_2 = 1.5, t_3 = 6.0$, and $t_4 = T = 80.0$ for CIFAR-10, and a 2-step inversion with $\varepsilon = t_1 = 0.07, t_2 = 1.8$, and $t_3 = T = 80.0$ for FFHQ and LSUN bedroom are sufficient for good reconstruction results. Note, that we use two different small initial noise, i.e., $\boldsymbol{\sigma}_1, \boldsymbol{\sigma}_2 \sim \mathcal{N}(0, \boldsymbol{I}), \boldsymbol{\sigma}_1 \neq \boldsymbol{\sigma}_2$ when inverting $\mathbf{x}_1$ and $\mathbf{x}_2$ respectively.

Since BCM learns to amplify the two initial Gaussian i.i.d. noises, it is reasonable to hypothesize that the amplified noises (i.e., the embeddings) $\mathbf{z}_1$ and $\mathbf{z}_2$ reside on the same hyperspherical surface as if $\mathbf{z}_1$ and $\mathbf{z}_2$ are directly sampled from $\mathcal{N}(0, T^2\boldsymbol{I})$. Therefore, following Song et al. (2023), we use spherical linear interpolation as

$$\mathbf{z} = \frac{\sin[(1-\alpha)\psi]}{\sin(\psi)}\mathbf{z}_1 + \frac{\sin[\alpha\psi]}{\sin(\psi)}\mathbf{z}_2, \tag{16}$$

in which $\alpha \in [0, 1]$ and $\psi = \arccos\left(\frac{\mathbf{z}_1^T \mathbf{z}_2}{\|\mathbf{z}_1\|_2 \|\mathbf{z}_2\|_2}\right)$.

As we discussed in the main text, it is crucial to set $\boldsymbol{\sigma}_1 \neq \boldsymbol{\sigma}_2$ for inversion. A possible reason is that if using $\boldsymbol{\sigma}_1 = \boldsymbol{\sigma}_2$ for inversion, the inverted noises $\mathbf{z}_1$ and $\mathbf{z}_2$ may reside on an unknown submanifold instead of the hyperspherical surface of Gaussian, and hence Equation (16) cannot yield ideal interpolation results. In Appendix F, we present visualization and discussions on the geometric properties of the noise space.

#### C.3.2 INPAINTING

We describe BCM's inpainting process in Algorithm 6. In our experiments on CIFAR-10, we set the initial noise scale $s = 0.5$. Additionally, instead of inverting the image back to $T = 80.0$, we empirically find inverting the image to $T = 2.0$ already suffice for satisfactory inpainting outcomes, and hence, we use a 3-step inversion (i.e., $N = 4$), where $t_1 = \varepsilon = 0.07$, $t_2 = 0.4$, $t_3 = 1.0$ and $t_4 = T = 2.0$, and 1-step generation from $T$ to 0.

As for the results by CMs, we implement CM's inpainting algorithm (Algorithm 4 in Song et al. (2023)). The total number of time steps (NFEs) is set to 18 according to the official inpainting script at `https://github.com/openai/consistency_models_cifar10/blob/main/editing_multistep_sampling.ipynb`. To ensure a fair comparison, we opt for the improved model, iCT (Song & Dhariwal, 2024), over the original CMs, since iCT delivers superior generation performance. Nonetheless, due to the absence of officially released codes and checkpoints by the authors, we reproduce and train our own iCT model. Our reproduced iCT yields an FID score of single-step generation closely matching that reported by Song & Dhariwal (2024) — our reproduced FID is 2.87 compared to the reported 2.83 — affirming the reliability of our results.

For higher-resolution images, we summarize BCM's inpainting with refinement in Algorithm 7. For a fair comparison among BCM's inpainting algorithm, CM's inpainting algorithm and BCM's inpainting with refinement, we run these three algorithms on the same BCM trained on FFHQ $64 \times 64$/LSUN Bedroom $256 \times 256$. Then, we perform the inpainting using the following hyperparameters:

- FFHQ $64 \times 64$: **BCM inpainting:** we adopt initial noise scale $s = 1.0$, 1-step inversion with $t_1 = \varepsilon = 0.5$ and $t_2 = T = 20.0$, and 1-step generation from $T$ to 0 in BCM's inpainting. **CM inpainting:** we adopt a 24-step noise schedule following Song et al. (2023). **BCM inpainting w. refine:** we first perform BCM inpainting, followed by refinement using only the last $M = 12$ time steps used in CM inpainting.

- LSUN Bedroom $256 \times 256$: **BCM inpainting:** we use initial noise scale $s = 0.5$, 2-step inversion with $t_1 = \varepsilon = 0.5, t_2 = 2.5$ and $t_3 = T = 200.0$, and 2-step ancestral generation with $t_1 = 4.4$ in BCM's inpainting. **CM inpainting:** we adopt a 40-step noise schedule following Song et al. (2023). we adopt the same 40-step noise schedule in Song et al. (2023). **BCM inpainting w. refine:** we first perform BCM inpainting, followed by refinement using only the last $M = 26$ time steps used in CM inpainting.

### C.3.3 JPEG RESTORATION

We invert the input image to reach the noise scale of $T = 3.2$ using $1/2/3$ steps and reconstruct it by single-step generation, making a total NFE of $2/3/4$ as in Figure 8. For 1-step inversion, we use $\varepsilon = t_1 = 0.5, t_2 = T = 3.2$. For 2-step inversion, we use $\varepsilon = t_1 = 0.5, t_2 = 1.2, t_3 = T = 3.2$. For 3-step inversion, we use $\varepsilon = t_1 = 0.5, t_2 = 1.2, t_3 = 2.0, t_4 = T = 3.2$.

For the I$^2$SB (Liu et al., 2023) baseline, since there's no pretrained checkpoints available for ImageNet-$64 \times 64$, we reproduce it using the official codebase at https://github.com/NVlabs/I2SB (NVIDIA Source Code License). There is also no official implemention or checkpoints available for Palette (Saharia et al., 2022), so we also reproduce it by modifying from the I$^2$SB codebase. We initialize both methods with a pretrained ADM model with weight available at https://github.com/openai/guided-diffusion (MIT License).

However, as there is no unconditional ADM weight for ImageNet-$64 \times 64$ and our BCM is also trained with class conditions, we *include labels in both the baseline methods and our method* for convenience and comparison fairness. This can lead to an overestimation of classifier accuracy or Inception Score (IS). Therefore, to fairly compare our method with the baselines, we choose to report MSE between the restored images and the ground truth, along with FID. This choice is motivated by the known rate-perception-distortion trade-off (Blau & Michaeli, 2018; 2019). In essence, for the same compression rate, there is a trade-off between MSE and realism, the latter of which is reflected by FID in our experiments. In our experiments, we found both our BCM and I$^2$SB models achieved a per pixel MSE of around 0.004. As for the FID, as shown in Figure 8, when NFE is limited, BCM achieves a better FID compared to I$^2$SB. This indicates that, *when NFE is limited, our BCM obtains a better restoration than I$^2$SB in a Pareto sense.*

## D DERIVATION OF THE NETWORK PARAMETERIZATION

In this section, we provide more details about our network parameterization design. To start with, recall that in Song et al. (2023), they parameterize the consistency model using skip connections as

$$\boldsymbol{f_\theta}(\mathbf{x}_t, t) = c_{\text{skip}}(t)\mathbf{x}_t + c_{\text{out}}(t)F_{\boldsymbol{\theta}}(c_{\text{in}}(t)\mathbf{x}_t, t), \tag{17}$$

in which

$$c_{\text{in}}(t) = \frac{1}{\sqrt{\sigma_{\text{data}}^2 + t^2}}, \quad c_{\text{out}}(t) = \frac{\sigma_{\text{data}}(t - \varepsilon)}{\sqrt{\sigma_{\text{data}}^2 + t^2}}, \quad c_{\text{skip}}(t) = \frac{\sigma_{\text{data}}^2}{\sigma_{\text{data}}^2 + (t - \varepsilon)^2}, \tag{18}$$

that ensures

$$c_{\text{out}}(\varepsilon) = 0, \quad c_{\text{skip}}(\varepsilon) = 1 \tag{19}$$

to hold at some very small noise scale $\varepsilon \approx 0$ so $\boldsymbol{f_\theta}(\mathbf{x}_0, \varepsilon) = \mathbf{x}_0$[4]. Since we expect the output is a noise image of target noise scale $u$, we expand the parameterization of $c_{\text{skip}}, c_{\text{out}}$ and $c_{\text{in}}$ to make them related to $u$, as

$$\boldsymbol{f_\theta}(\mathbf{x}_t, t, u) = c_{\text{skip}}(t, u)\mathbf{x}_t + c_{\text{out}}(t, u)F_{\boldsymbol{\theta}}(c_{\text{in}}(t, u)\mathbf{x}_t, t, u). \tag{20}$$

Our derivation of network parameterization shares the same group of principles in EDM Karras et al. (2022). Specifically, we first require the input to the network $F_{\boldsymbol{\theta}}$ to have unit variance. Following Eq. (114) $\sim$ (117) in EDM paper (Karras et al., 2022), we have

$$c_{\text{in}}(t, u) = \frac{1}{\sqrt{\sigma_{\text{data}}^2 + t^2}}. \tag{21}$$

Then, as we discussed in the main text, we expect the model to achieve consistency in Equation (7) along the entire trajectory. According to Lemma 1 in Song et al. (2023), we have

$$\nabla \log p_t(\mathbf{x}_t) = \frac{1}{t^2} \left( \mathbb{E}[\mathbf{x}|\mathbf{x}_t] - \mathbf{x}_t \right) \tag{22}$$

$$\overset{(i)}{\approx} \frac{1}{t^2} \left( \mathbf{x} - \mathbf{x}_t \right) \tag{23}$$

$$= -\frac{\boldsymbol{\sigma}}{t}, \tag{24}$$

in which we follow Song et al. (2023) to estimate the expectation with $\mathbf{x}$ in $(i)$. When $u$ and $t$ are close, we can use the Euler solver to estimate $\mathbf{x}_u$, i.e,

$$\mathbf{x}_u \approx \mathbf{x}_t - t(u - t)\nabla \log p_t(\mathbf{x}_t) \tag{25}$$

$$= \mathbf{x}_t + (u - t)\boldsymbol{\sigma} \tag{26}$$

$$= \mathbf{x} + u\boldsymbol{\sigma}. \tag{27}$$

Therefore, when $u$ and $t$ are close, and base on Song et al. (2023)'s approximation in $(i)$, we can rewrite the consistency defined in Equation (7) as

$$d\left(\boldsymbol{f_\theta}(\mathbf{x} + t\boldsymbol{\sigma}, t, u), \mathbf{x} + u\boldsymbol{\sigma}\right) \tag{28}$$

$$= |\boldsymbol{f_\theta}(\mathbf{x} + t\boldsymbol{\sigma}, t, u) - (\mathbf{x} + u\boldsymbol{\sigma})| \tag{29}$$

$$= |c_{\text{skip}}(t, u)(\mathbf{x} + t\boldsymbol{\sigma}) + c_{\text{out}}(t, u)F_{\boldsymbol{\theta}}(c_{\text{in}}(t, u)(\mathbf{x} + t\boldsymbol{\sigma}), t, u) - (\mathbf{x} + u\boldsymbol{\sigma})| \tag{30}$$

$$= |c_{\text{out}}(t, u)F_{\boldsymbol{\theta}}(c_{\text{in}}(t, u)(\mathbf{x} + t\boldsymbol{\sigma}), t, u) - (\mathbf{x} + u\boldsymbol{\sigma} - c_{\text{skip}}(t, u)(\mathbf{x} + t\boldsymbol{\sigma}))| \tag{31}$$

$$= |c_{\text{out}}(t, u)| \cdot \left| F_{\boldsymbol{\theta}}(c_{\text{in}}(t, u)(\mathbf{x} + t\boldsymbol{\sigma}), t, u) \right.$$
$$\left. - \frac{1}{c_{\text{out}}(t, u)} \left((1 - c_{\text{skip}}(t, u))\mathbf{x} + (u - c_{\text{skip}}(t, u)t)\boldsymbol{\sigma}\right) \right|. \tag{32}$$

For simplicity, we set $d(\cdot, \cdot)$ to $L_1$ norm in Equation (29). Note that, in practice, for $D$-dimensional data, we follow Song & Dhariwal (2024) to use Pseudo-Huber loss $d(\mathbf{a}, \mathbf{b}) = \sqrt{||\mathbf{a} - \mathbf{b}||^2 + 0.00054^2 D} - 0.00054\sqrt{D}$, which can be well approximated by $L_1$ norm.

We should note that Equation (28) is based on the assumption that $u$ and $t$ are reasonably close. *This derivation is only for the pursuit of reasonable parameterization and should not directly serve as an objective function.* Instead, one should use the soft constraint we proposed in Equation (9) as the objective function.

The approximate effective training target of network $F_{\boldsymbol{\theta}}$ is therefore

$$\frac{1}{c_{\text{out}}(t, u)} \left((1 - c_{\text{skip}}(t, u))\mathbf{x} + (u - c_{\text{skip}}(t, u)t)\boldsymbol{\sigma}\right). \tag{33}$$

---

[4]While the parameterization written in the original paper of Song et al. (2023) did not explicitly include $c_{\text{in}}(t)$, we find it is actually included in its official implementation at https://github.com/openai/consistency_models_cifar10/blob/main/jcm/models/utils.py#L189 in the form of Equations (17) and (18).

Following Karras et al. (2022), we require the effective training target to have unit variance, i.e.,

$$\text{Var}\left[\frac{1}{c_{\text{out}}(t,u)}\left((1-c_{\text{skip}}(t,u))\mathbf{x}+(u-c_{\text{skip}}(t,u)t)\boldsymbol{\sigma}\right)\right]=1, \tag{34}$$

so we have

$$c_{\text{out}}^2(t,u)=\text{Var}\left[(1-c_{\text{skip}}(t,u))\mathbf{x}+(u-c_{\text{skip}}(t,u)t)\boldsymbol{\sigma}\right] \tag{35}$$

$$=(1-c_{\text{skip}}(t,u))^2\sigma_{\text{data}}^2+(u-c_{\text{skip}}(t,u)t)^2 \tag{36}$$

$$=(\sigma_{\text{data}}^2+t^2)c_{\text{skip}}^2(t,u)-2(\sigma_{\text{data}}^2+tu)c_{\text{skip}}(t,u)+(\sigma_{\text{data}}^2+u^2), \tag{37}$$

which is a hyperbolic function of $c_{\text{skip}}(t,u)$. Following Karras et al. (2022), we select $c_{\text{skip}}(t,u)$ to minimize $|c_{\text{out}}(t,u)|$ so that the errors of $F_{\boldsymbol{\theta}}$ are amplified as little as possible, as

$$c_{\text{skip}}(t,u)=\underset{c_{\text{skip}}(t,u)}{\arg\min}|c_{\text{out}}(t,u)|=\underset{c_{\text{skip}}(t,u)}{\arg\min}c_{\text{out}}^2(t,u). \tag{38}$$

So we have

$$(\sigma_{\text{data}}^2+t^2)c_{\text{skip}}(t,u)=\sigma_{\text{data}}^2+tu \tag{39}$$

$$c_{\text{skip}}(t,u)=\frac{\sigma_{\text{data}}^2+tu}{\sigma_{\text{data}}^2+t^2}. \tag{40}$$

Substituting Equation (40) into Equation (36), we have

$$c_{\text{out}}^2(t,u)=\frac{\sigma_{\text{data}}^2t^2(t-u)^2}{\left(\sigma_{\text{data}}^2+t^2\right)^2}+\left(\frac{\sigma_{\text{data}}^2t+t^2u}{\sigma_{\text{data}}^2+t^2}-u\right)^2 \tag{41}$$

$$=\frac{\sigma_{\text{data}}^2t^2(t-u)^2+\sigma_{\text{data}}^4(t-u)^2}{\left(\sigma_{\text{data}}^2+t^2\right)^2} \tag{42}$$

$$=\frac{\sigma_{\text{data}}^2(t-u)^2}{\sigma_{\text{data}}^2+t^2}, \tag{43}$$

and finally

$$c_{\text{out}}(t,u)=\frac{\sigma_{\text{data}}(t-u)}{\sqrt{\sigma_{\text{data}}^2+t^2}}. \tag{44}$$

One can immediately verify that when $u=t$, $c_{\text{skip}}(t,u)=1$ and $c_{\text{out}}(t,u)=0$ so that the boundary condition

$$\boldsymbol{f_\theta}(\mathbf{x}_t,t,t)=\mathbf{x}_t \tag{45}$$

holds.

On the side of CMs, setting $u=\varepsilon$ will arrive at exactly the same form of $c_{\text{in}}(t,u)$ and $c_{\text{out}}(t,u)$ in Equation (18). While $c_{\text{skip}}(t,u)$ does not degenerate exactly to the form in Equation (18) when

taking $u = \varepsilon$ and $t > \varepsilon$, this inconsistency is negligible when $\varepsilon \approx 0$, as

$$\left|c_{\text{skip}}^{\text{BCM}}(t, \varepsilon) - c_{\text{skip}}^{\text{CM}}(t)\right| = \left|\frac{\sigma_{\text{data}}^2 + t\varepsilon}{\sigma_{\text{data}}^2 + t^2} - \frac{\sigma_{\text{data}}^2}{\sigma_{\text{data}}^2 + (t - \varepsilon)^2}\right| \tag{46}$$

$$= \frac{\left|\left(\sigma_{\text{data}}^2 + (t - \varepsilon)^2\right)\left(\sigma_{\text{data}}^2 + t\varepsilon\right) - \sigma_{\text{data}}^2\left(\sigma_{\text{data}}^2 + t^2\right)\right|}{\left(\sigma_{\text{data}}^2 + t^2\right)\left(\sigma_{\text{data}}^2 + (t - \varepsilon)^2\right)} \tag{47}$$

$$= \frac{\left|\varepsilon^2\sigma_{\text{data}}^2 - \varepsilon t\sigma_{\text{data}}^2 + \varepsilon t(t - \varepsilon)^2\right|}{\left(\sigma_{\text{data}}^2 + t^2\right)\left(\sigma_{\text{data}}^2 + (t - \varepsilon)^2\right)} \tag{48}$$

$$= \frac{\varepsilon(t - \varepsilon)\left|(t - \varepsilon)t - \sigma_{\text{data}}^2\right|}{\left(\sigma_{\text{data}}^2 + t^2\right)\left(\sigma_{\text{data}}^2 + (t - \varepsilon)^2\right)} \tag{49}$$

$$< \frac{\varepsilon(t - \varepsilon)\max\left\{(t - \varepsilon)t, \sigma_{\text{data}}^2\right\}}{\left(\sigma_{\text{data}}^2 + t^2\right)\left(\sigma_{\text{data}}^2 + (t - \varepsilon)^2\right)} \tag{50}$$

$$\leqslant \frac{\varepsilon(t - \varepsilon)\max\left\{t^2, \sigma_{\text{data}}^2\right\}}{\left(\sigma_{\text{data}}^2 + t^2\right)\left(\sigma_{\text{data}}^2 + (t - \varepsilon)^2\right)} \tag{51}$$

$$< \frac{\varepsilon(t - \varepsilon)}{\sigma_{\text{data}}^2 + (t - \varepsilon)^2} \tag{52}$$

$$= \frac{\varepsilon}{\frac{\sigma_{\text{data}}^2}{t - \varepsilon} + (t - \varepsilon)} \tag{53}$$

$$\leqslant \frac{\varepsilon}{2\sigma_{\text{data}}}. \tag{54}$$

Therefore, we conclude that our parameterization is compatible with CM's parameterization, so with the same CT target of Equation (6), any CT techniques (Song et al., 2023; Song & Dhariwal, 2024) should directly apply to our model and it should inherit all properties from CMs just by setting $u = \varepsilon$, which is a clear advantage compared with models that adopt completely different parameterizations (e.g., CTM Kim et al. (2024)).

# E    COMPARISON OF CT, CTM AND BCT

We compare the training objective of CT, CTM and our proposed BCT in Table 2, where we can see how our method naturally extends CT and differs from CTM.

# F    UNDERSTANDING THE LEARNED NOISE ("EMBEDDING") SPACE

This section provides some insights into the learned "embedding" space. Recall that during inversion (Algorithm 5), we first inject a small Gaussian noise to the image. Here we investigate the influence of this noise and the original image content on the noise generated by inversion.

We randomly select 500 CIFAR-10 images, and randomly split them into 10 groups. We then invert the images to their corresponding noise by Algorithm 5. We inject the *same* initial noise during inversion for images in the same group. Figure 15 visualize the t-SNE results Van der Maaten & Hinton (2008) of the inversion outcomes. In Figure 15a, images injected with the same initial noise are shown in the same color; while in Figure 15b, we color the points according to their class label (i.e., airplane, bird, cat, ...).

Interestingly, we can see that images inverted with the same initial noise are clustered together. We, therefore, conjecture that each initial noise corresponds to a submanifold in the final "embedding" space. The union of all these submanifolds constitutes the final "embedding" space, which is the typical set of $\mathcal{N}(0, T^2\boldsymbol{I})$, closed to a hypersphere. This explains why applying the same initial noise is suboptimal in interpolation, as discussed in Appendix C.3.1.

Table 2: Comparison of CT, CTM training, and BCT training methodology. The figures illustrate the main objective of each method, where $\bar{\theta}$ stands for stop gradient operation. Note that for BCM, there are two possible scenarios corresponding to the denoising and diffusion direction, respectively.

| Model | Illustration of Training Objective | Detailed Form of Loss |
|---|---|---|
| CT | | $\mathcal{L}_{CT} = \mathbb{E}_{t_n,\mathbf{x}}[\lambda(t_n)d],$ 

 $\mathbf{x}$ is the training sample, 
 $\lambda(\cdot)$ is the reweighting function, 
 $t_n, d$ are illustrated in the left plot. |
| CTM | | $\mathcal{L}_{CTM} = \mathbb{E}_{t_n,t_{n'},t_{n''},\mathbf{x}}[d]$ 
 $\quad + \lambda_{GAN}\mathcal{L}_{GAN} + \lambda_{DSM}\mathcal{L}_{DSM}$ 

 $\mathcal{L}_{DSM}$ is the adversarial loss, 
 $\mathcal{L}_{DSM}$ is Denoising Score Matching loss (Song et al., 2021b; Vincent, 2011), 
 $\lambda_{GAN}, \lambda_{DSM}$ are the reweighting functions, 
 $t_n, t_{n'}, t_{n''}, d$ are illustrated in the left plot. |
| BCT | | $\mathcal{L}_{BCT} = \mathbb{E}_{t_n,t_{n'},\mathbf{x}}[\lambda(t_n)d_1 + \lambda'(t_n,t_{n'})d_2],$ 

 $\lambda(\cdot), \lambda'(\cdot,\cdot)$ are the reweighting functions, 
 $t_n, t_{n'}, d_1, d_2$ are illustrated in the left plot. |

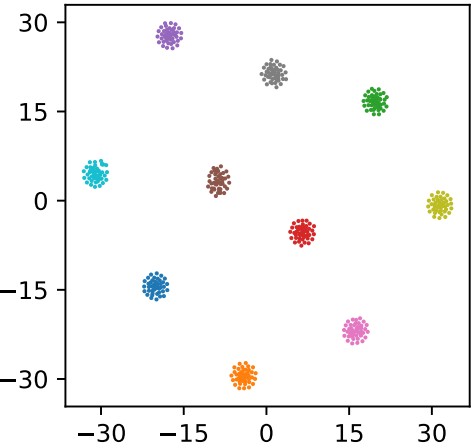

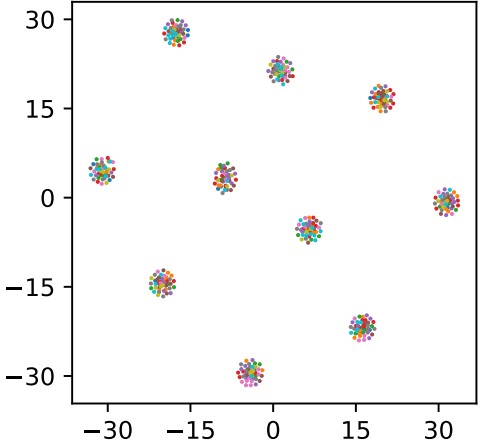

(a) images inverted with the same initial noise are shown in the same color

(b) images with the same class label are shown in the same color

Figure 15: t-SNE of the inverted noise generated from 500 randomly selected CIFAR images.

## G ADDITIONAL VISUALIZATIONS

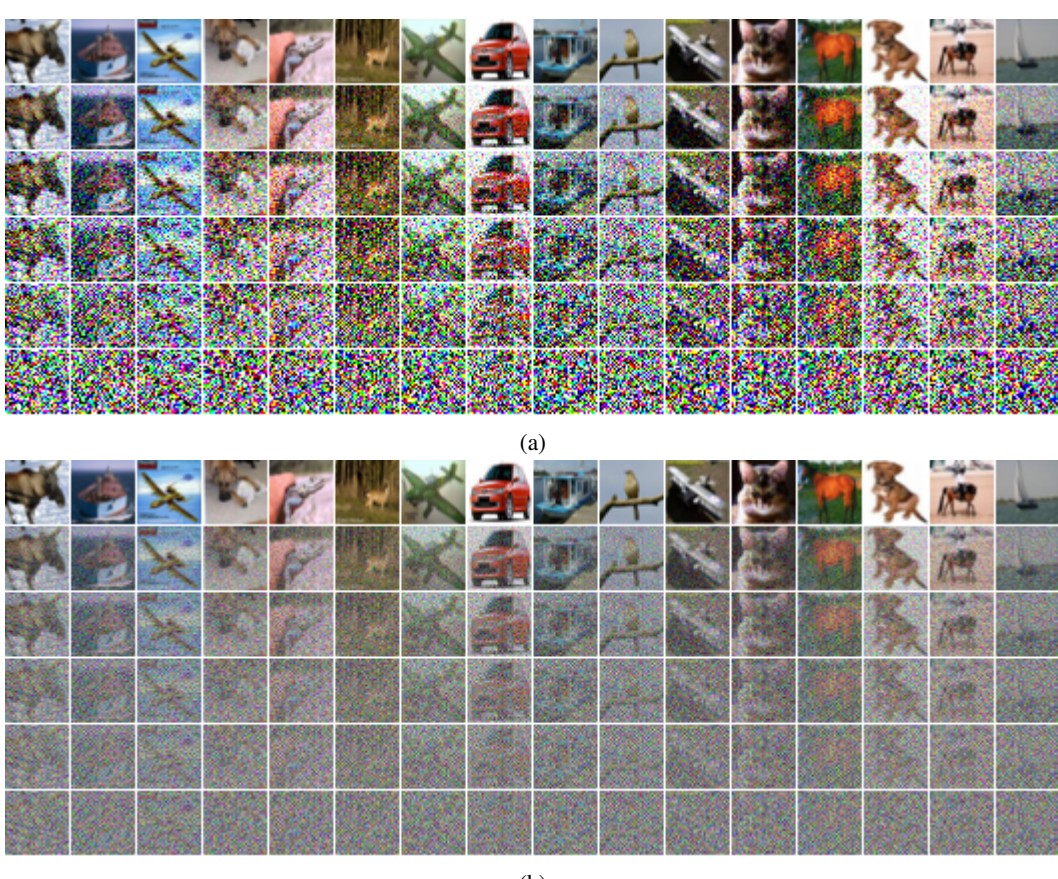

(a)

(b)

Figure 16: Visualization of the noise images generated by inversion with BCM. Each line corresponds to a noise scale of $0, 0.2, 0.5, 1.0, 2.0, 80.0$, respectively. In (a), we truncate the image to $[-1, 1]$, while in (b) we normalize the image to $[-1, 1]$.

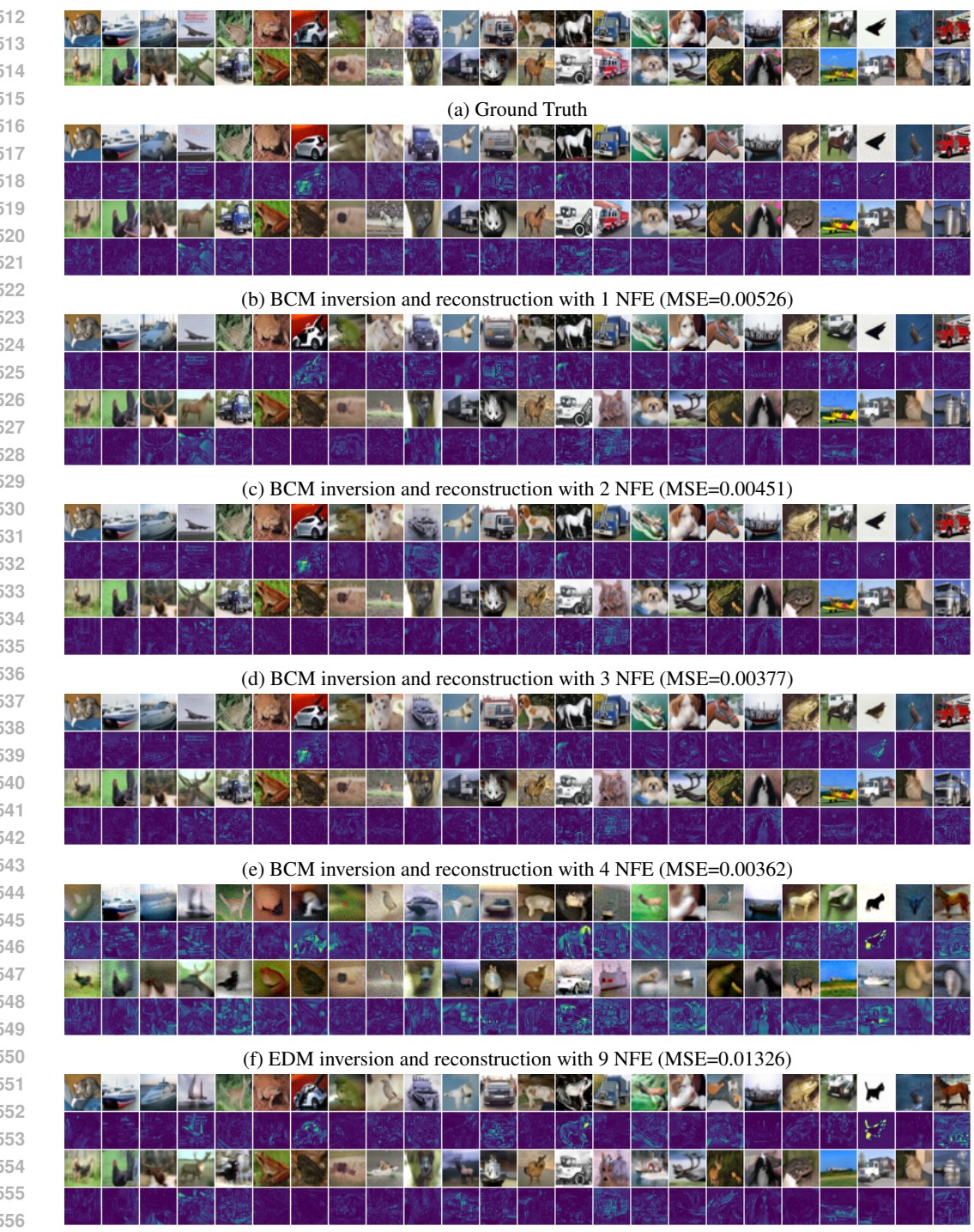

(a) Ground Truth

(b) BCM inversion and reconstruction with 1 NFE (MSE=0.00526)

(c) BCM inversion and reconstruction with 2 NFE (MSE=0.00451)

(d) BCM inversion and reconstruction with 3 NFE (MSE=0.00377)

(e) BCM inversion and reconstruction with 4 NFE (MSE=0.00362)

(f) EDM inversion and reconstruction with 9 NFE (MSE=0.01326)

(g) EDM inversion and reconstruction with 19 NFE (MSE=0.00421)

Figure 17: Reconstructed images and their residual with unconditional BCM on CIFAR-10. We include EDM's results in (e) and (f) for comparison.

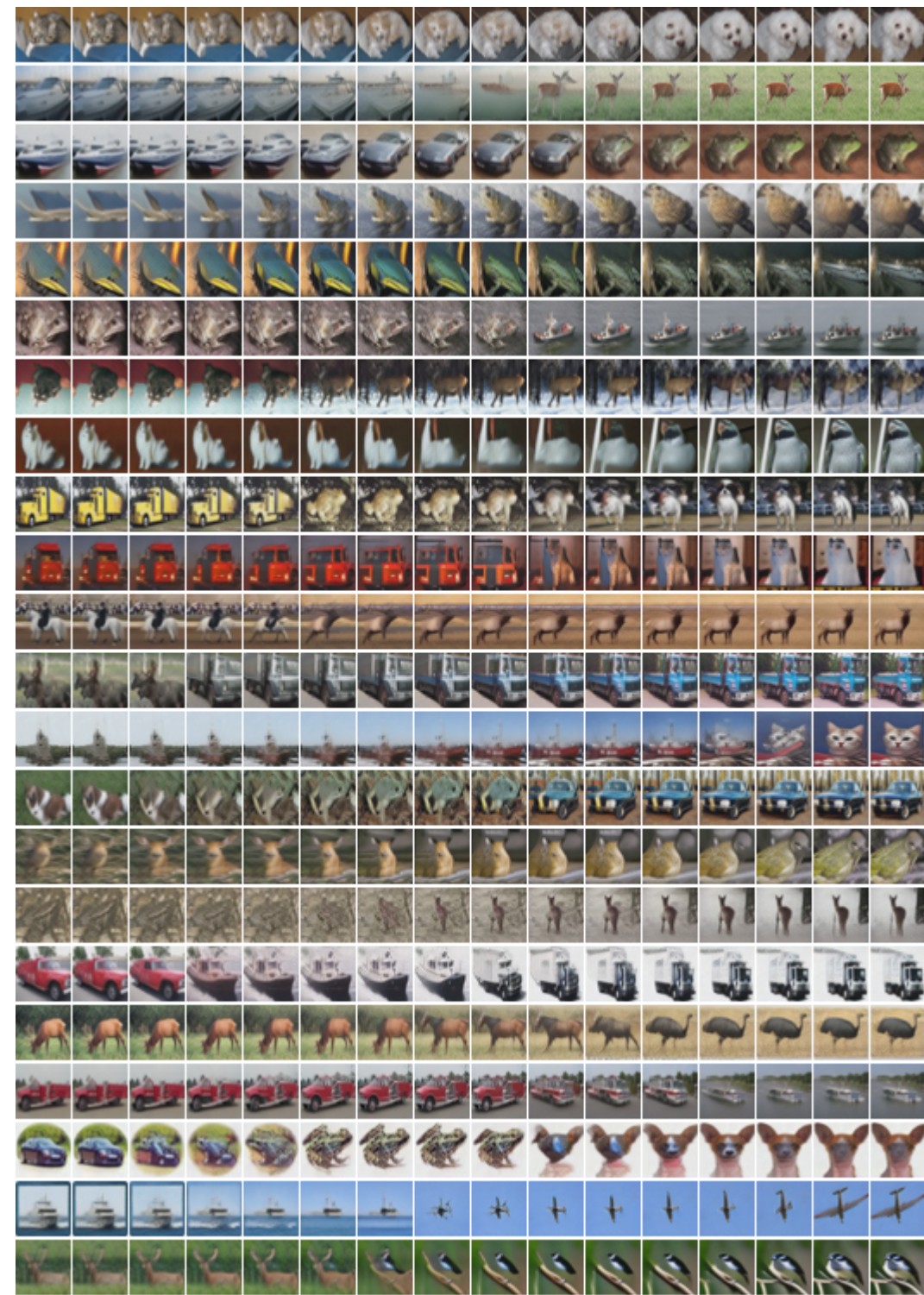

Figure 18: Interpolation between two real CIFAR-10 images (injecting *different* initial noise in inversion).

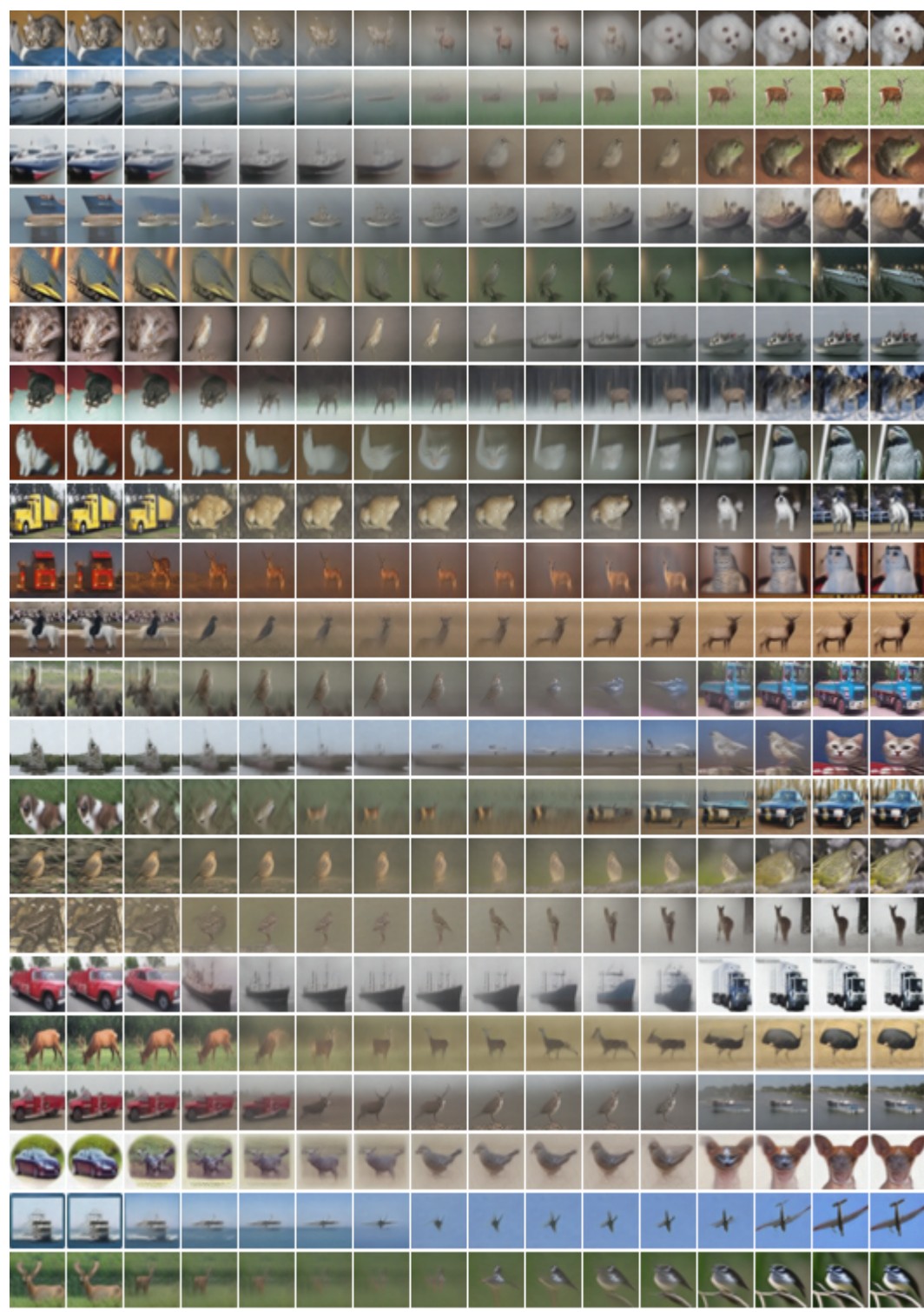

Figure 19: Interpolation between two real CIFAR-10 images (injecting the *same* initial noise in inversion).

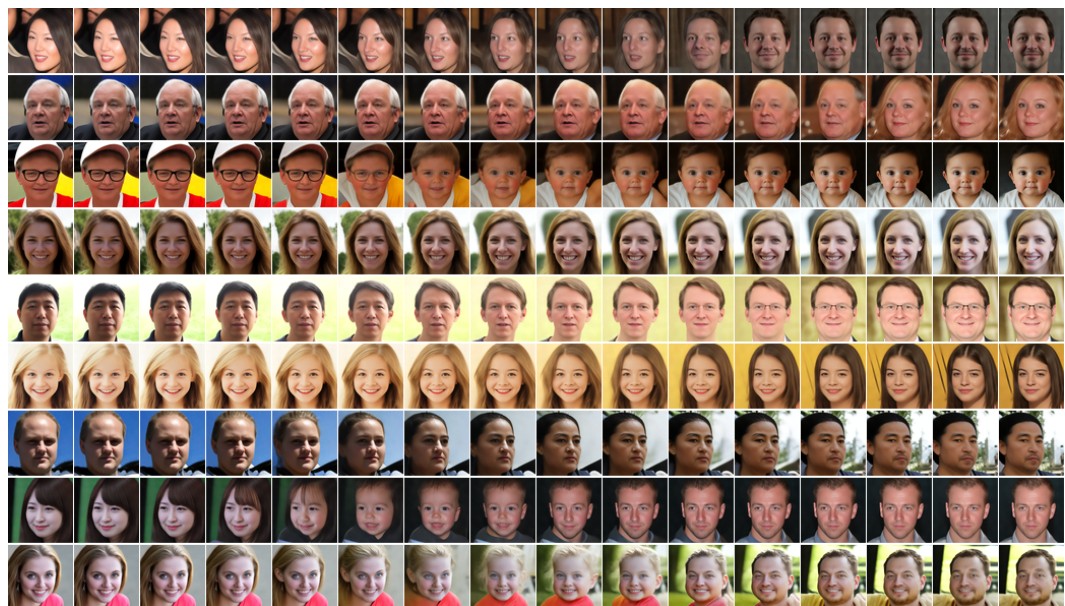

Figure 20: Interpolation between two real FFHQ $64 \times 64$ images (injecting *different* initial noise in inversion).

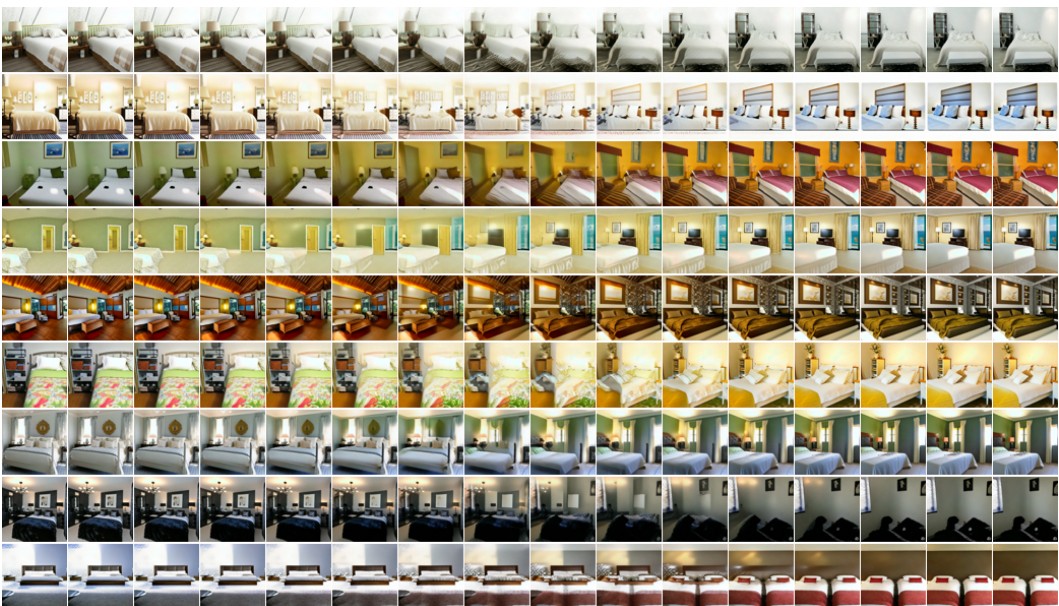

Figure 21: Interpolation between two real LSUN Bedroom $256 \times 256$ images (injecting *different* initial noise in inversion).

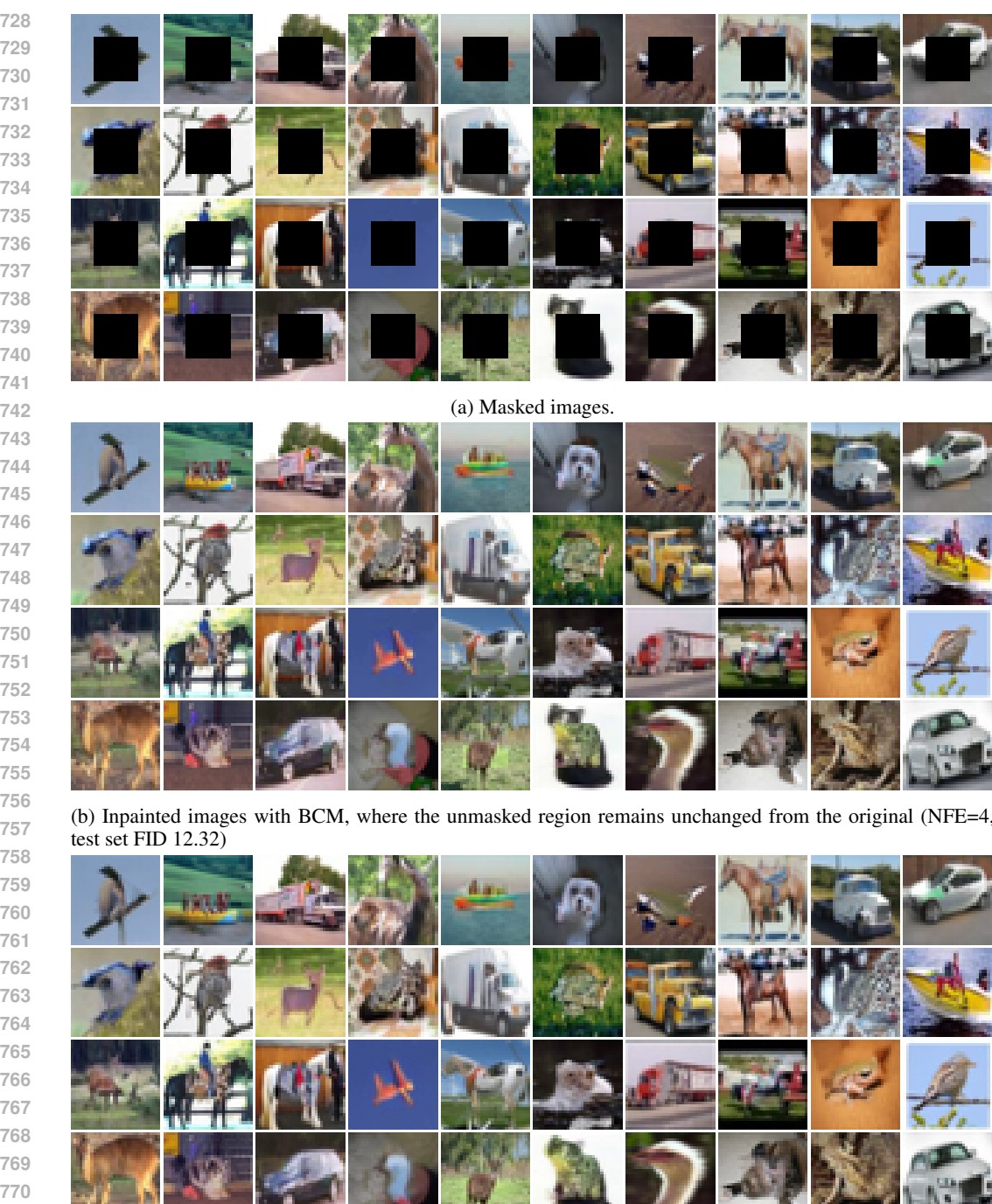

(a) Masked images.

(b) Inpainted images with BCM, where the unmasked region remains unchanged from the original (NFE=4, test set FID 12.32)

(c) Inpainted images with BCM, with feathered masks for seamless integration.

Figure 22: Inpainting with BCM on CIFAR-10.

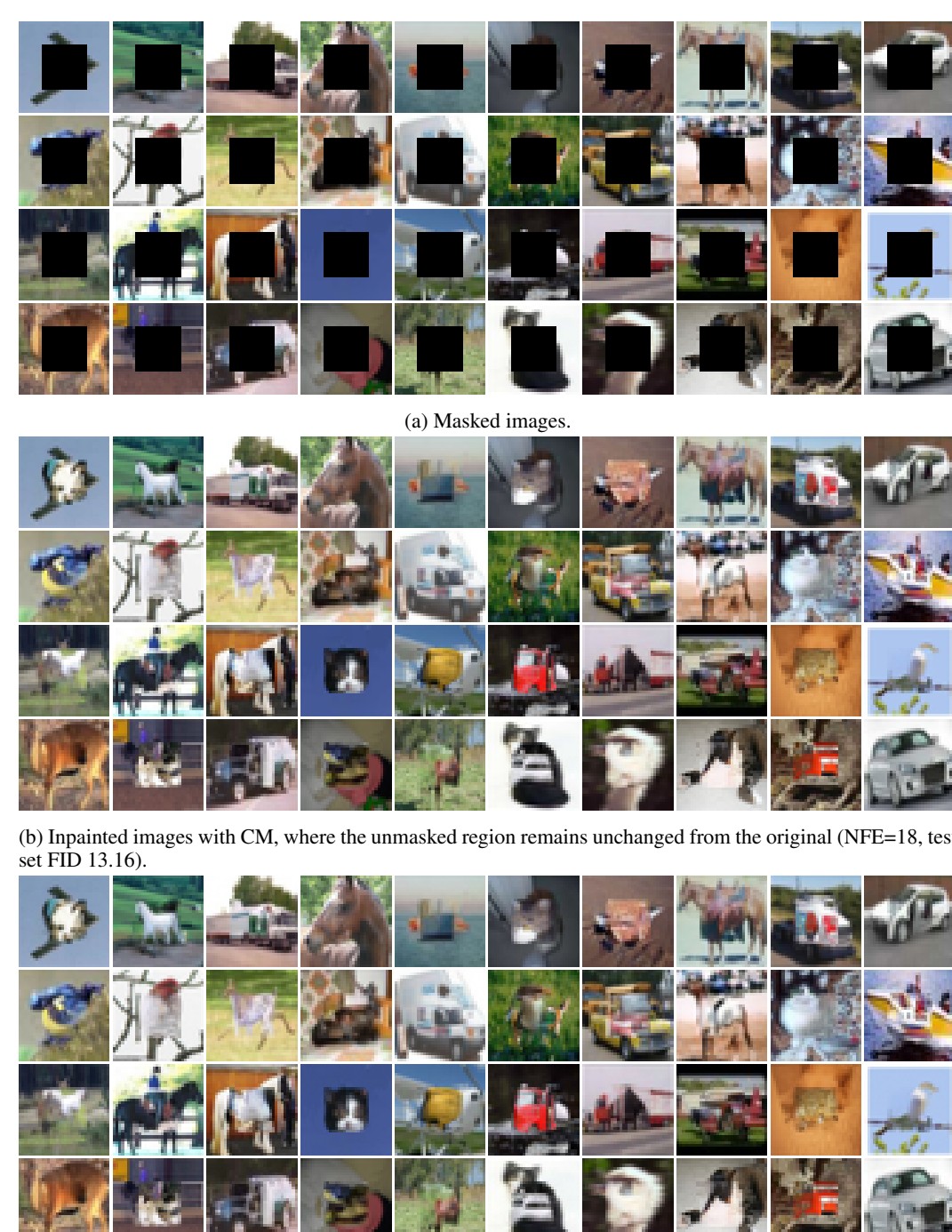

(a) Masked images.

(b) Inpainted images with CM, where the unmasked region remains unchanged from the original (NFE=18, test set FID 13.16).

(c) Inpainted images with CM, with feathered masks for seamless integration.

Figure 23: Inpainting with CM (concretely, iCT) on CIFAR-10.

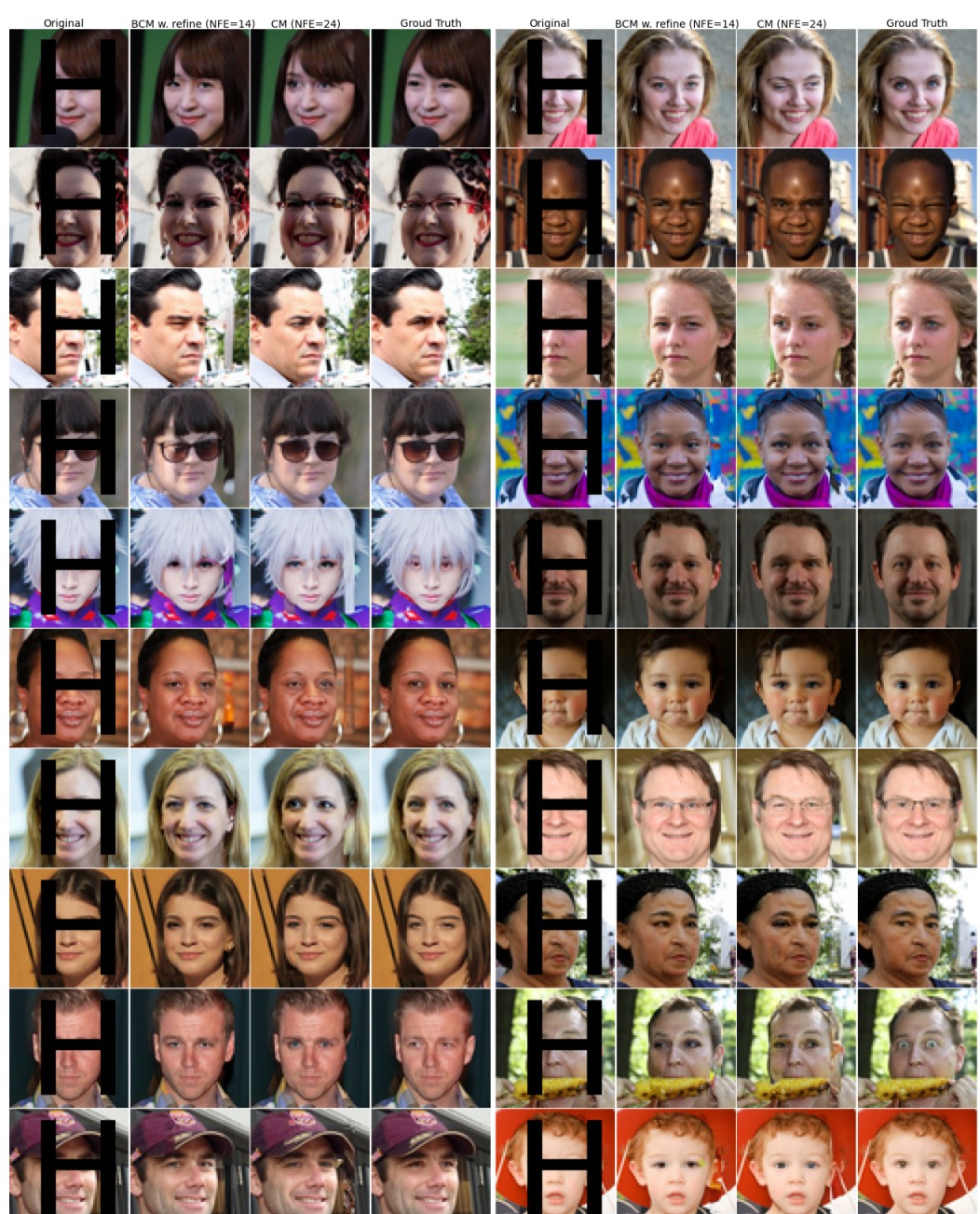

Figure 24: Inpainting on FFHQ $64 \times 64$.

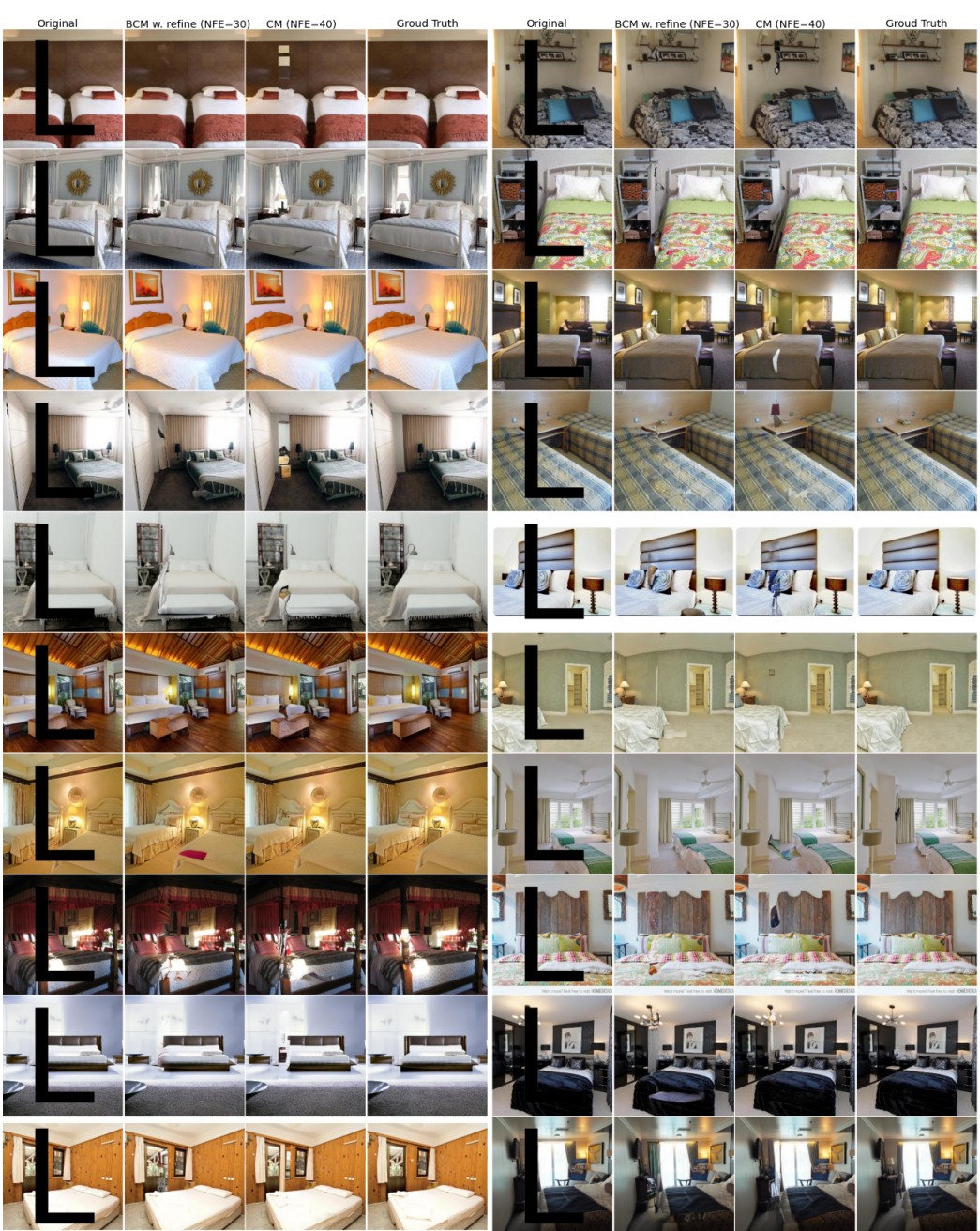

Figure 25: Inpainting on LSUN Bedroom $256 \times 256$.

## H MORE GENERATION SAMPLES

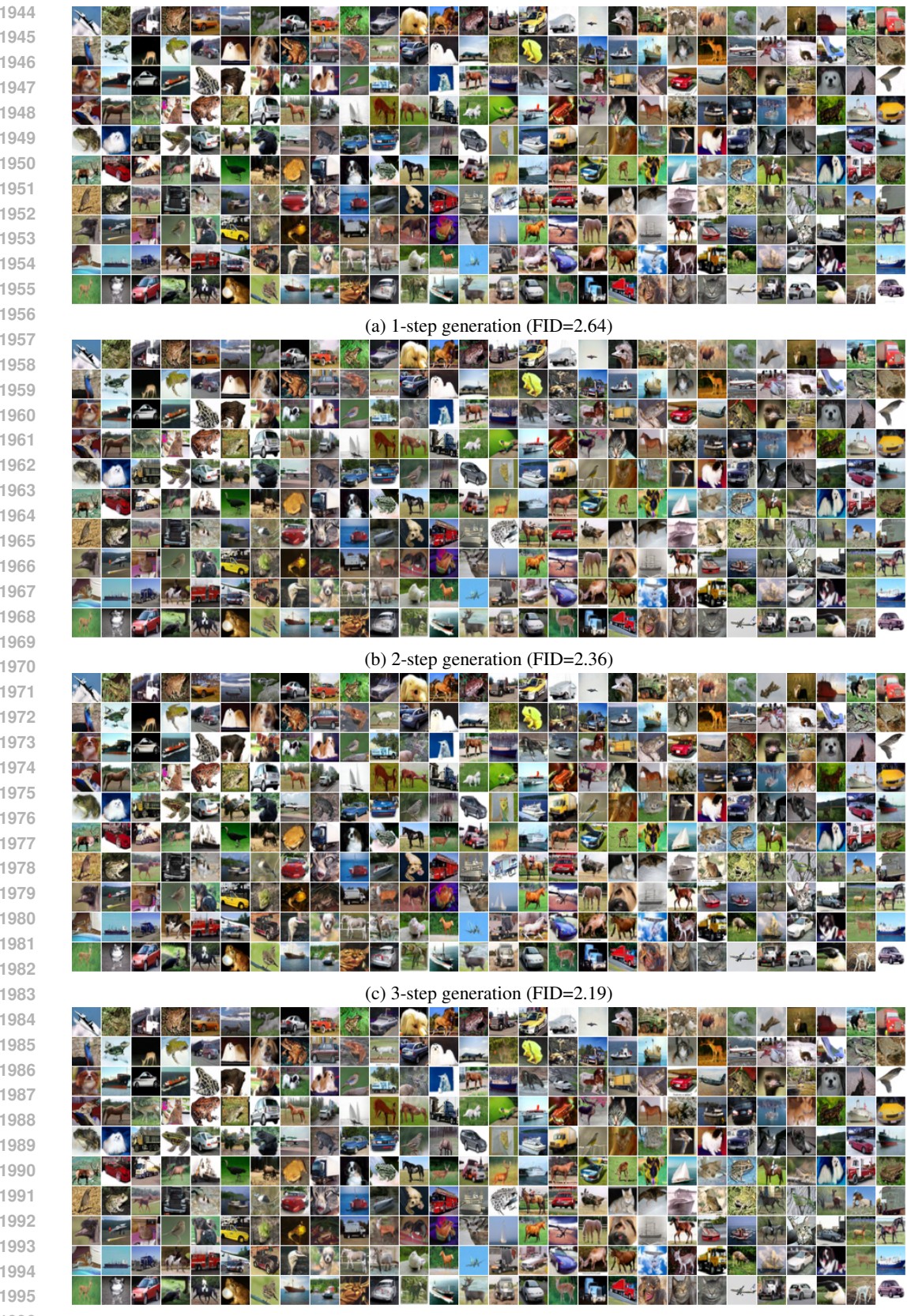

(a) 1-step generation (FID=2.64)

(b) 2-step generation (FID=2.36)

(c) 3-step generation (FID=2.19)

(d) 4-step generation (FID=2.07)

Figure 26: Uncurated CIFAR-10 samples generated by BCM-deep.

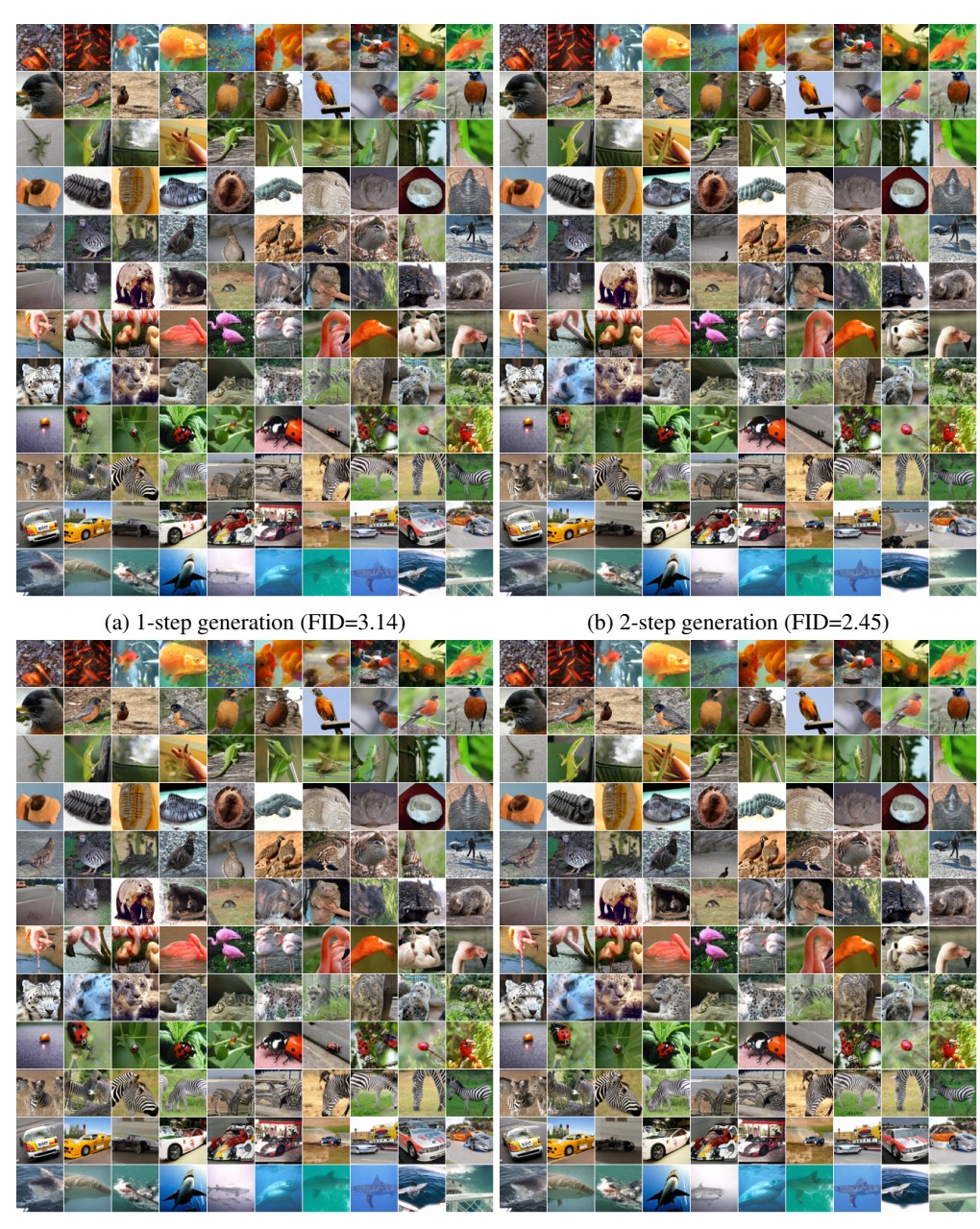

(a) 1-step generation (FID=3.14)   (b) 2-step generation (FID=2.45)

(c) 3-step generation (FID=2.61)   (d) 4-step generation (FID=2.35)

Figure 27: Uncurated ImageNet-64 samples generated by BCM-deep. Each line corresponds to one randomly selected class.

