# OpenReview forum: "Bidirectional Consistency Models"
_ICLR.cc/2025/Conference — ICLR 2025 Conference Withdrawn Submission_

### Official Review · Reviewer_LwcN · 2024-10-21

**Soundness:** 3
**Presentation:** 3
**Contribution:** 3
**Rating:** 5
**Confidence:** 4

**Summary:**

Following Consistency Models (CM - Song 2023) and Consistency Trajectory Models (CTM - Kim 2024), this paper offers their generalization - a learned network that enables going in both directions of the trajectory of the ODE solution path. More specifically, while CM enables a jump from any t to t=0, and CTM from any t to any u<t, the proposed BCM is designed to remove the u<t constraint, thereby enabling moving up and down the trajectory. The bidirectionality is obtained by desiginbing a specific loss that generalizes that of CTM, and training the network accordingly, either by starting from scratch or by leveraging an existing pretrained CM machine.

The main benefits (as claimed) in this new capability are editing and recovery abilities, such as image interpolation, inpainting, and JPEG restoration. The cost is a slight loss in synthesis performance, as the network is now more challenged.

**Strengths:**

1. This is a well written paper and pleasant to read.
2. The authors worked hard on the various experiments presented.
3. The proposed method seems to work very well, with a slight cost in the quality of image synthesis.

**Weaknesses:**

1. The novelty in composing the generalization offered is mild at best (the change in the loss), as the method is very similar to CM and CTM.
2. As the key benefits of the proposed approach are the ability to invert the generation process and leverage it, I find several flaws/weaknesses in the proposed experiments and results:
a. The inversion error is not negligible, and without much control over its magnitude. Could something be done to reduce this error using more NFE's?
b. In JPEG restoration, it is not enough to show FID versus NFE as Figure 8 does. Are the obtained images consistent with the given compressd image in PSNR terms? i.e. if we recompress the image using the same Quality factor, will we get close enough to the starting image? This performance measure (CPSNR) should be reported. Also, example image results should be included.
c. On the same JPEG topic - there are other techniques that are far better than PALETTE that should have been used in the comparisons (e.g. https://openreview.net/forum?id=O3WJOt79289, DPS: https://openreview.net/forum?id=OnD9zGAGT0k, PiGDM: https://openreview.net/forum?id=9_gsMA8MRKQ).
d. The inpainting results are far from satisfactory, as evident esspecialy in Figure 9. If 14 NFE are necessary, a comparison to methods beyond CM should be provided, in the spirit of the comment above - DPS, PiGDM and DDRM (https://ddrm-ml.github.io/) should be compared. Indeed, rich work in the past 3 years offers inpaiting techniques via diffusion and these should be mentioned and compared with. See [1-12] below.
e. To summarize, if the key benefit of thsi work is the reverse diretion that CM and CTM do not provide, this work is failing to convience of the benefits in this new capability. See my later comment on editing - perhaps this should have been the main application outlet to this work.

[1] Zehao Dou and Yang Song. Diffusion posterior sampling for linear inverse problem solving: A filtering perspective. In The Twelfth International Conference on Learning Representations, 2024.
[2] Berthy T. Feng, Jamie Smith, Michael Rubinstein, Huiwen Chang, Katherine L. Bouman, and William T. Freeman. Score-based diffusion models as principled priors for inverse imaging. In Proceedings of the IEEE/CVF International Conference on Computer Vision (ICCV), pp. 10520–10531, October 2023.
[3] Gabriel Cardoso, Yazid Janati el idrissi, Sylvain Le Corff, and Eric Moulines. Monte carlo guided denoising diffusion models for bayesian linear inverse problems. In The Twelfth International Conference on Learning Representations, 2024.
[4] Hyungjin Chung, Jeongsol Kim, Michael Thompson Mccann, Marc Louis Klasky, and Jong Chul Ye. Diffusion posterior sampling for general noisy inverse problems. In The Eleventh International Conference on Learning Representations, 2023.
[5] Ciprian Corneanu, Raghudeep Gadde, and Aleix M. Martinez. Latentpaint: Image inpainting in latent space with diffusion models. In Proceedings of the IEEE/CVF Winter Conference on Applications of Computer Vision (WACV), pp. 4334–4343, January 2024.
[6] Anji Liu, Mathias Niepert, and Guy Van den Broeck. Image inpainting via tractable steering of diffusion models. In The Twelfth International Conference on Learning Representations, 2024.
[7] Andreas Lugmayr, Martin Danelljan, Andr´es Romero, Fisher Yu, Radu Timofte, and Luc Van Gool. Repaint: Inpainting using denoising diffusion probabilistic models. 2022 IEEE/CVF Conference on Computer Vision and Pattern Recognition (CVPR), pp. 11451–11461, 2022.
[8] Litu Rout, Negin Raoof, Giannis Daras, Constantine Caramanis, Alex Dimakis, and Sanjay Shakkottai. Solving linear inverse problems provably via posterior sampling with latent diffusion models. In Thirty-seventh Conference on Neural Information Processing Systems, 2023.
[9] Bowen Song, Soo Min Kwon, Zecheng Zhang, Xinyu Hu, Qing Qu, and Liyue Shen. Solving inverse problems with latent diffusion models via hard data consistency. In The Twelfth International Conference on Learning Representations, 2024.
[10] Yinhuai Wang, Jiwen Yu, and Jian Zhang. Zero-shot image restoration using denoising diffusion null-space model. The Eleventh International Conference on Learning Representations, 2023.
[11] Guanhua Zhang, Jiabao Ji, Yang Zhang, Mo Yu, Tommi S. Jaakkola, and Shiyu Chang. Towards coherent image inpainting using denoising diffusion implicit models. CoRR, abs/2304.03322, 2023.
[12] Morteza Mardani, Jiaming Song, Jan Kautz, and Arash Vahdat. A variational perspective on solving inverse problems with diffusion models. In The Twelfth International Conference on Learning Representations, 2024.

**Questions:**

If inversion is so effective with this method, can it be incorporated into editing algorithms for fine-tuning an image? e.g. LEDITS++? This should be discussed.

I will be happy to raise the rating of the paper if I get satisfactory answers to all the above issues.

**Details Of Ethics Concerns:**

None.

---

### Official Review · Reviewer_Vt2e · 2024-10-30

**Soundness:** 3
**Presentation:** 4
**Contribution:** 2
**Rating:** 6
**Confidence:** 3

**Summary:**

This paper present a new model based on diffusion named bidirectional consistency models (BCM). It learns the forward and the backward PF ODE of diffusion wth the same model. It allows to develop new sampling strategy including zigzag sampling where noise are removed and amplified with the BCM. The authors show experimentally that BCM outperforms other methods with the same NFE for image generation or image restoration.

**Strengths:**

This paper is well written and progressive. BCM shows outstanding experimental performance. The exact method including hyperparameters choice is explained in details in Appendix and the choice of reparametrization is well motivated. The comparison with recent state of the art method is well explained.

**Weaknesses:**

Major weakness:
- It is not clear why there is a need to learn the forward PF ODE. In fact, adding noise to an image is easy. Therefore, it is not clear what is the gain of BCM compare to CM.
- No theoretical analysis are provided. It is not clear why the sampling strategy of BCM will effectively sample the right distribution.

Minor weakness:
- I suggest in table 1 to separate more clearly each method. Because some methods are runned with various NFE, I suggest to separate with a horizental line each method.

**Questions:**

- What is the idea of the exponential Moving average (EMA) in line 140 ? Is the optimization algorithm to minimize (3) only converging in average ?
- In Algorithm (1), the BCT seems to be a SGD algorithm to minimize $\mathcal{L}_{BCT}$, to your point of view, what will be the gain of using other stochastic algorithms of order 1 such as RMSProp or ADAM to minimize this loss ?
- What is the gain of using the Pseudo-Huber loss instead of the $L_1$ norm if they are close as stated in line 1283 ? Could you provide any theoretical advantages of Pseudo-Huber loss in this specific context ?
- Can you explain what is the stop gradient operation ? (line 135 and line 219)
- On Figure 10, the loss of BCT seems to decrease with piece wise constant, how to you interprete that behavior ?
- In the zigzag strategy, a random noise is added then amplified with the network. Why applyfing random noise with the network is different from adding random noise with the right scale imediately ? Could you provide quantitative empirical evidence that demonstrate any advantage of amplifying the noise instead of adding larger noise (Figure 2 gaves a qualitative comparison) ? Could you explain theoretically the different between these two strategies ?
- In table 1, most results are taken from other papers, how do you ensure that these results are comparable ? Especially, these methods are not using the same network. Can you include the number of parameter in the table ?
- What are the theoretical guarantees that the BCM sampling strategies, with all empirical approximations (neural network), effectively sample the desire distribution ?

---

### Official Review · Reviewer_J5mp · 2024-11-07

**Soundness:** 2
**Presentation:** 2
**Contribution:** 2
**Rating:** 5
**Confidence:** 5

**Summary:**

The paper proposes a model called Bidirectional Consistency Model (BCM), which aims to simultaneously optimize image generation and inverse mapping (i.e., noise reduction) tasks. The design of BCM enables the model to perform bidirectional generation and inversion operations on the probability flow ODE (PF ODE) trajectory by introducing bidirectional consistency. The traditional diffusion model (DM) and consistency model (CM) have the problems of slow generation and unstable reverse processes. BCM introduces a single neural network to complete the generation and reverse tasks, thus significantly improving the computational efficiency and performance.

**Strengths:**

1. BCM implements one-step generation and inversion operations, greatly reducing the amount of computation required.
2. The model is suitable for a variety of tasks such as interpolation, restoration, and image compression recovery, and has a wide range of application potential.

**Weaknesses:**

The experiment section should be improved. Please refer the detailed comment below.

**Questions:**

1. The experiments in this paper primarily focus on CIFAR-10 and ImageNet-64, which are relatively small-sized datasets. Testing the proposed method on larger, more complex datasets (e.g., ImageNet-256) could better validate its scalability and practical performance in real-world applications.

2. While the paper claims that BCM significantly reduces the number of function evaluations (NFE), it would be beneficial to provide a direct comparison of inference time against other models, particularly consistency models and diffusion models with fast samplers.

3.  It is interesting to note that in Table 1, the FID score for BCM worsens when the NFE increases from 2 to 3. Could you provide more details?

4. For the inpainting task, the paper only applies a box mask on CIFAR-10, leaving out tests on other datasets with this mask.

5. Given BCM’s demonstrated performance on inpainting and JPEG artifact removal, it would be valuable to investigate its effectiveness on other image restoration tasks, such as super-resolution.

6. Figure 8 provides quantitative FID scores for JPEG restoration but lacks visual examples of the restored images.

---

### Official Review · Reviewer_avd1 · 2024-11-10

**Soundness:** 2
**Presentation:** 3
**Contribution:** 4
**Rating:** 5
**Confidence:** 3

**Summary:**

The paper introduces Bidirectional Consistency Models (BCM), which extend the capabilities of consistency models by enabling an inversion process. Building on the previously proposed consistency training loss, the authors add an additional loss term to ensure the bidirectional property between any two-time steps. Beyond training, the paper also proposes several novel sampling methods specific to BCM. Experimental results demonstrate that BCM performs comparably to previous methods in image generation while unlocking new functionalities, such as solving inverse problems. These enhanced capabilities arise from BCM’s inversion property.

**Strengths:**

1. The method proposed in this paper shows strong potential. Nearly all diffusion model-based image editing techniques require inversion, and previous methods have largely depended on DDIM and DDIM Inversion, which typically require hundreds of function evaluations (NFE). To my knowledge, BCM may be the first method to reduce NFE to fewer than ten. This advancement suggests that BCM could have a significant impact on applications requiring diffusion model inversion.

2. The paper’s writing is clear, with a logical organization that is easy to follow. The detailed descriptions provided make the method straightforward to reproduce.

**Weaknesses:**

1. The training loss proposed in Equation 10 and the sampling method outlined in Section 3.3 are primarily derived from an intuitive perspective. Compared to previous works on consistency models and consistency trajectory models, this paper lacks some theoretical analysis regarding convergence.
2. While the paper includes sufficient experiments to validate its methods, it does not provide enough evidence to emphasize the bidirectional property, especially in terms of application. In Section 4.3, for the interpolation results, I think the difference between interpolating generated images and real images doesn't fully demonstrate the strength of the bidirectional property. Similarly, inpainting is also achievable with the consistency model, it does not showcase the unique advantage of bidirectionality. Also, both inpainting and blind restoration of compressed images could be approached as inverse problems, solving without inversion. Thus, none of the experiments in Section 4.3 clearly illustrate the distinct benefits of bidirectionality. Additionally, Section 4.1 focuses on image generation, which is not directly related to inversion. As a result, only Section 4.2 directly supports BCM’s efficiency in inversion.

I think the paper could contain some applications that cannot achieved without inversion, such as image editing. Otherwise, why do others choose BCM, as iCM and CTM have better generation performance?

**Questions:**

1. In line 252, why is the reweighting parameter $\lambda'(t_n, t_n')$ defined in this way, and how does it help maintain the loss scale?

2. In both the Zigzag sampling and the inversion process, the authors introduce a small noise term, explaining that the endpoint of the time horizon is less effectively covered. This explanation, however, is somewhat confusing. If the endpoint at $t = 0$ is indeed less effectively covered, why is it necessary to return to $t = 0$ during sampling? Additionally, do the authors compare the effects of different sampling endpoints, such as $t = 0$ versus $t = 0.07$ (mentioned in line 334)?

**Details Of Ethics Concerns:**

No ethics review is needed.

---

### Official Review · Reviewer_YY7U · 2024-11-12

**Soundness:** 3
**Presentation:** 3
**Contribution:** 3
**Rating:** 6
**Confidence:** 3

**Summary:**

This submission focuses on consistency models, addressing the challenge of bidirectionality by ensuring these models are invertible, allowing a path from data back to noise. To achieve this, the paper proposes Bidirectional Consistency Models (BCM), which leverage a single neural network for both generation and inversion. This design enables one-step generation and inversion. For low NFE settings, the experiments demonstrate that BCM outperforms both I2SB and Palette in a fully zero-shot manner.

**Strengths:**

- Addressing bidirectionality in consistency models is a timely and impactful problem, contributing to the field's advancement in invertibility and versatility.

- The approach is conceptually simple, making it accessible and logical.

- Experimental results convincingly demonstrate that BCM achieves notable gains compared to CM and CTM, underscoring its potential.

**Weaknesses:**

The methods section could benefit from clearer writing; given the simplicity of the core idea, a more structured and precise explanation would improve readability.

**Questions:**

- Typos: Notably, there is a typo in the abstract (“wh ich”).

- Time Sampling and Discretization: It would be beneficial to comment on whether uniform sampling is an optimal choice. Drawing from practices in EDM, it could be advantageous to sample the pair (u,t)(u, t)(u,t) primarily from the mid-trajectory region, potentially with a center-concentrated distribution, to capture more informative states along the trajectory.

- Inpainting Experiments: Instead of initializing with a masked image filled with Gaussian noise within the masked regions, an alternative approach might be to retain zero-masking, invert to identify the noise, interpolate in the noise space, and then generate. Performing these manipulations directly in the latent noise space might offer more interpretability and flexibility for different tasks. This aligns with the interpretative benefits of the noise space and is supported by ideas found in [1].

[1] Chang P, Tang J, Gross M, Azevedo VC. How I Warped Your Noise: a Temporally-Correlated Noise Prior for Diffusion Models. InThe Twelfth International Conference on Learning Representations 2024.

---

### Note · Authors · 2024-11-27

**Comment:**

We thank the reviewers for their detailed feedback. We will incorporate their suggestions to enhance our manuscript.

**Withdrawal Confirmation:**

I have read and agree with the venue's withdrawal policy on behalf of myself and my co-authors.